# Simple Drop-in LoRA Conditioning on Attention Layers Will Improve Your Diffusion Model

**Joo Young Choi**     *lthilnklover@snu.ac.kr*
*Department of Mathematical Sciences*
*Seoul National University*

**Jaesung R. Park**     *ryanpark7@snu.ac.kr*
*Department of Mathematical Sciences*
*Seoul National University*

**Inkyu Park**     *inkyupark@krafton.com*
*KRAFTON*

**Jaewoong Cho**     *jwcho@krafton.com*
*KRAFTON*

**Albert No**     *albertno@yonsei.ac.kr*
*Department of Artificial Intelligence*
*Yonsei University*

**Ernest K. Ryu**     *eryu@math.ucla.edu*
*Department of Mathematics*
*University of California, Los Angeles*

**Reviewed on OpenReview:** *https://openreview.net/forum?id=38P40gJPrI*

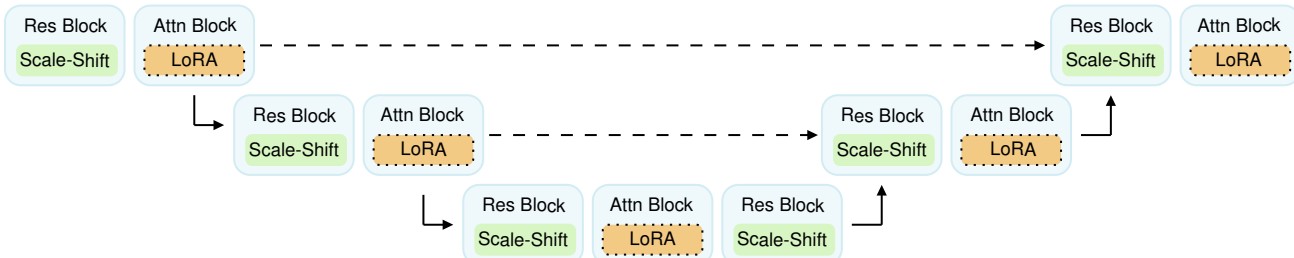

Figure 1: The standard U-Net architecture for diffusion models conditions convolutional layers in residual blocks with scale-and-shift but does not condition attention blocks. Simply adding **LoRA** conditioning on attention layers improves the image generation quality.

## Abstract

Current state-of-the-art diffusion models employ U-Net architectures containing convolutional and (qkv) self-attention layers. The U-Net processes images while being conditioned on the time embedding input for each sampling step and the class or caption embedding input corresponding to the desired conditional generation. Such conditioning involves scale-and-shift operations to the convolutional layers but does not directly affect the attention layers. While these standard architectural choices are certainly effective, not conditioning the attention layers feels arbitrary and potentially suboptimal. In this work, we show that simply adding LoRA conditioning to the attention layers without changing or tuning the other parts of the U-Net architecture improves the image generation quality. For example, a drop-in addition of LoRA conditioning to EDM diffusion model yields FID scores of 1.91/1.75 for unconditional and class-conditional CIFAR-10 generation, improving upon the baseline of 1.97/1.79. Code is available at `https://github.com/lthilnklover/diffusion_lora`.

| Dataset | Model | Type | NFE | # basis | rank | FID↓ | # Params |
|---|---|---|---|---|---|---|---|
| CIFAR-10 (uncond.) | IDDPM | baseline | 4000 | - | - | 3.69 | 52546438 |
| | | only LoRA | 4001 | 11 | 4 | 3.64 | **47591440** |
| | | with LoRA | 4001 | 11 | 4 | **3.37** | 54880528 |
| | EDM (vp) | baseline | 35 | - | - | 1.97 | 55733891 |
| | | only LoRA | 35 | 18 | 4 | 1.99 | **53411675** |
| | | with LoRA | 35 | 18 | 4 | **1.96/1.91** | 57745499 |
| CIFAR-10 (cond.) | IDDPM | baseline | 4000 | - | - | 3.38 | 52551558 |
| | | only LoRA | 4001 | (11, 10) | 4 | **2.91** | 48513040 |
| | | with LoRA | 4001 | (11, 10) | 4 | 3.12 | 55807248 |
| | EDM (vp) | baseline | 35 | - | - | 1.79 | 55735299 |
| | | only LoRA | 35 | 18 | 4 | 1.82 | **53413083** |
| | | with LoRA | 35 | 18 | 4 | **1.75** | 57746907 |
| ImageNet64 (uncond.) | IDDPM | baseline | 4000 | - | - | 19.2 | 121063942 |
| | | only LoRA | 4001 | 11 | 4 | 18.2 | **113602960** |
| | | with LoRA | 4001 | 11 | 4 | **18.1** | 139278224 |
| FFHQ64 (uncond.) | EDM (vp) | baseline | 79 | - | - | 2.39 | 61805571 |
| | | only LoRA | 79 | 20 | 4 | 2.46 | **58935795** |
| | | with LoRA | 79 | 20 | 4 | **2.37/2.28** | 63941631 |

Table 1: Image generation results. We compare three different conditionings for each setting: **(baseline)** conditioning only convolutional layers in residual blocks with scale-and-shift; **(only LoRA)** conditioning only attention layers using LoRA conditioning and not conditioning convolutional layers; **(with LoRA)** conditioning both convolutional layers and attention layers with scale-and-shift and LoRA conditioning, respectively. For unconditional CIFAR-10 and FFHQ64 sampling using EDM with LoRA, we also report the FID score obtained by initializing the base model with pre-trained weights.

# 1 Introduction

In recent years, diffusion models have led to phenomenal advancements in image generation. Many cutting-edge diffusion models leverage U-Net architectures as their backbone, consisting of convolutional and (qkv) self-attention layers (Dhariwal & Nichol, 2021; Kim et al., 2023; Saharia et al., 2022; Rombach et al., 2022; Podell et al., 2024). In these models, the U-Net architecture-based score network is conditioned on the time, and/or, class, text embedding (Ho & Salimans, 2021) using scale-and-shift operations applied to the convolutional layers in the so-called residual blocks. Notably, however, the attention layers are not directly affected by the conditioning, and the rationale behind not extending conditioning to attention layers remains unclear. This gap suggests a need for in-depth studies searching for effective conditioning methods for attention layers and assessing their impact on performance.

Meanwhile, low-rank adaptation (LoRA) has become the standard approach for parameter-efficient fine-tuning of large language models (LLM) (Hu et al., 2022). With LoRA, one trains low-rank updates that are added to frozen pre-trained dense weights in the attention layers of LLMs. The consistent effectiveness of LoRA for LLMs suggests that LoRA may be generally compatible with attention layers used in different architectures and for different tasks (Chen et al., 2022; Pan et al., 2022; Lin et al., 2023; Gong et al., 2024).

In this work, we introduce a novel method for effectively conditioning the attention layers in the U-Net architectures of diffusion models by jointly training multiple LoRA adapters along with the base model. We call these LoRA adapters TimeLoRA and ClassLoRA for discrete-time settings, and Unified Compositional LoRA (UC-LoRA) for continuous signal-to-ratio (SNR) settings. Simply adding these LoRA adapters in a drop-in fashion *without modifying or tuning the configurations of original model* brings consistent enhancement in FID scores across several popular models applied to CIFAR-10, FFHQ 64x64, and ImageNet datasets. In particular, adding LoRA-conditioning to the EDM model (Karras et al., 2022) yields improved FID scores of 1.75, 1.91, 2.28 for class-conditional CIFAR-10, unconditional CIFAR-10, and FFHQ 64x64 datasets, respectively, outperforming the baseline scores of 1.79, 1.97, 2.39. Moreover, we find that LoRA conditioning

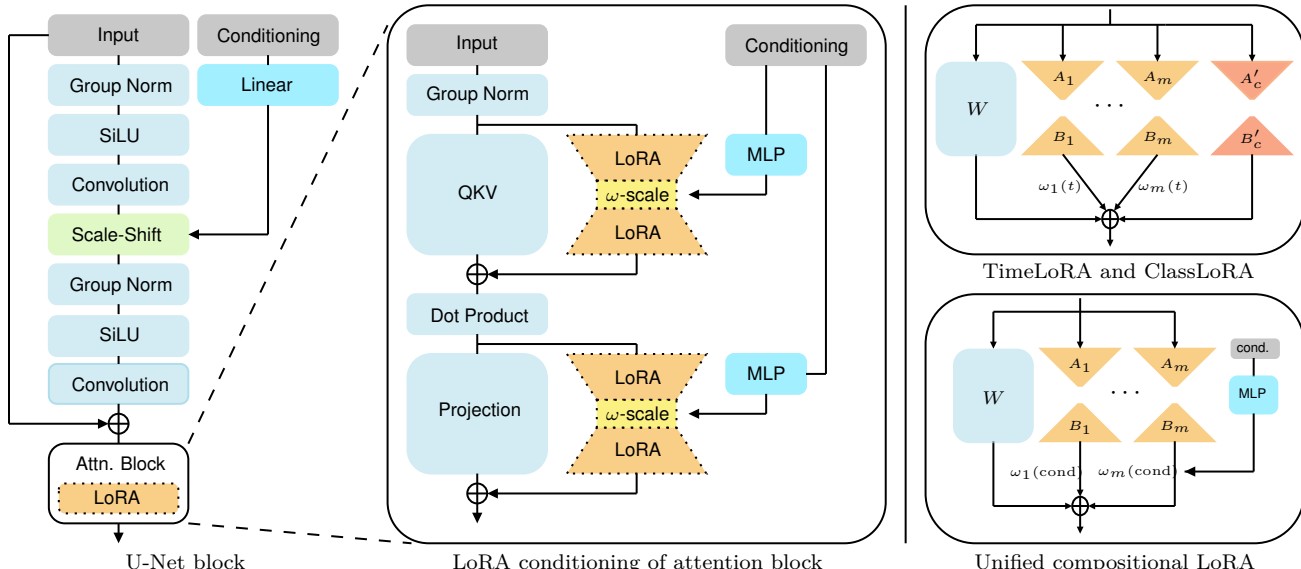

Figure 2: Conditioning of U-Net Block: **(left)** scale-and-shift conditioning on the convolutional block **(middle)** LoRA conditioning on the attention block **(right)** *top*: TimeLoRA and ClassLoRA for the discrete-time setting, *bottom*: unified composition LoRA for the continuous-SNR setting.

by itself is powerful enough to perform effectively. Our experiments show that only conditioning the attention layers using LoRA adapters (without the conditioning convolutional layers with scale-and-shift) achieves comparable FID scores compared to the baseline scale-and-shift conditioning (without LoRA).

**Contribution.** Our experiments show that using LoRA to condition time and class information on attention layers is effective across various models and datasets, including nano diffusion (Lelarge et al., 2024), IDDPM (Nichol & Dhariwal, 2021), and EDM (Karras et al., 2022) architectures using the MNIST (Deng, 2012), CIFAR-10 (Krizhevsky et al., 2009), and FFHQ (Karras et al., 2019) datasets.

Our main contributions are as follows. (i) We show that simple drop-in LoRA conditioning on the attention layers improves the image generation quality, as measured by lower FID scores, while incurring minimal ($\sim$10%) added memory and compute costs. (ii) We identify the problem of whether to and how to condition attention layers in diffusion models and provide the positive answer that attention layers should be conditioned and LoRA is an effective approach that outperforms the prior approaches of no conditioning or conditioning with adaLN (Peebles & Xie, 2023).

Our results advocate for incorporating LoRA conditioning into the larger state-of-the-art U-Net-based diffusion models and the newer experimental architectures.

## 2   Prior work and preliminaries

### 2.1   Diffusion models

Diffusion models (Sohl-Dickstein et al., 2015; Song & Ermon, 2019; Ho et al., 2020; Song et al., 2021b) generate images by iteratively removing noise from a noisy image. This denoising process is defined by the reverse process of the forward diffusion process: given data $x_0 \sim q_0$, progressively inject noise to $x_0$ by

$$q(x_t \,|\, x_{t-1}) = \mathcal{N}\left(\sqrt{1 - \beta_t}x_{t-1}, \beta_t\mathbf{I}\right)$$

for $t = 1, \ldots, T$ and $0 < \beta_t < 1$. If $\beta_t$ is sufficiently small, we can approximate the reverse process as

$$q(x_{t-1} \,|\, x_t) \approx \mathcal{N}\left(\mu_t(x_t), \beta_t\mathbf{I}\right)$$

where

$$\mu_t(x_t) = \frac{1}{\sqrt{1-\beta_t}}(x_t + \beta_t \nabla \log p_t(x_t)).$$

A diffusion model is trained to approximate the *score function* $\nabla \log p_t(x_t)$ with a *score network* $s_\theta$, which is often modeled with a U-Net architecture (Ronneberger et al., 2015; Song & Ermon, 2019). With $s_\theta \approx \nabla \log p_t(x_t)$, the diffusion model approximates the reverse process as

$$p_\theta(x_{t-1}|x_t) = \mathcal{N}\left(\frac{1}{\sqrt{1-\beta_t}}(x_t + \beta_t s_\theta(x_t, t)), \beta_t \mathbf{I}\right)$$
$$\approx q(x_{t-1} \,|\, x_t).$$

To sample from a trained diffusion model, one starts with Gaussian noise $x_T \sim \mathcal{N}(0, (1-\bar{\alpha}_T)\mathbf{I})$, where $\bar{\alpha}_t = \prod_{s=1}^{t}(1-\beta_s)$, and progressively denoise the image by sampling from $p_\theta(x_{t-1}|x_t)$ with $t = T, T-1, \dots, 2, 1$ sequentially to obtain a clean image $x_0$.

The above discrete-time description of diffusion models has a continuous-time counterpart based on the theory of stochastic differential equation (SDE) for the forward-corruption process and reversing it based on Anderson's reverse-time SDE (Anderson, 1982) or a reverse-time ordinary differential equation (ODE) with equivalent marginal probabilities (Song et al., 2021a). Higher-order integrators have been used to reduce the discretization errors in solving the differential equations (Karras et al., 2022).

**Architecture for diffusion models.** The initial work of Song & Ermon (2019) first utilized the CNN-based U-Net architecture (Ronneberger et al., 2015) as the architecture for the score network. Several improvements have been made by later works (Ho et al., 2020; Nichol & Dhariwal, 2021; Dhariwal & Nichol, 2021; Hoogeboom et al., 2023) incorporating multi-head self-attention (Vaswani et al., 2017), group normalization (Wu & He, 2018), and adaptive layer normalization (adaLN) (Perez et al., 2018). Recently, several alternative architectures have been proposed. Jabri et al. (2023) proposed Recurrent Interface Network (RIN), which decouples the core computation and the dimension of the data for more scalable image generation. Peebles & Xie (2023); Bao et al. (2023); Gao et al. (2023); Hatamizadeh et al. (2023) investigated the effectiveness of transformer-based architectures (Dosovitskiy et al., 2021) for diffusion models. Yan et al. (2023) utilized state space models (Gu et al., 2022) in DiffuSSM to present an attention-free diffusion model architecture. In this work, we propose a conditioning method for attention layers and test it on several CNN-based U-Net architectures. Note that our proposed method is applicable to all diffusion models utilizing attention layers.

## 2.2 Low-rank adaptation

Using trainable adapters for specific tasks has been an effective approach for fine-tuning models in the realm of natural language processing (NLP) (Houlsby et al., 2019; Pfeiffer et al., 2020). Low-rank adpatation (LoRA, Hu et al. (2022)) is a parameter-efficient fine-tuning method that updates a low-rank adapter: to fine-tune a pre-trained dense weight matrix $W \in \mathbb{R}^{d_{\text{out}} \times d_{\text{in}}}$, LoRA parameterizes the fine-tuning update $\Delta W$ with a low-rank factorization

$$W + \Delta W = W + BA,$$

where $B \in \mathbb{R}^{d_{\text{out}} \times r}, A \in \mathbb{R}^{r \times d_{\text{in}}}$, and $r \ll \min\{d_{\text{in}}, d_{\text{out}}\}$.

**LoRA and diffusion.** Although initially proposed for fine-tuning LLMs, LoRA is generally applicable to a wide range of other deep-learning modalities. Recent works used LoRA with diffusion models for various tasks including image generation (Ryu, 2023; Gu et al., 2023; Go et al., 2023), image editing (Shi et al., 2023), continual learning (Smith et al., 2023), and distillation (Golnari, 2023; Wang et al., 2023b). While all these works demonstrate the flexibility and efficacy of the LoRA architecture used for fine-tuning diffusion models, to the best of our knowledge, our work is the first attempt to use LoRA as part of the core U-Net for diffusion models for full training, not fine-tuning.

### 2.3 Conditioning the score network

For diffusion models to work properly, it is crucial that the score network $s_\theta$ is conditioned on appropriate side information. In the base formulation, the score function $\nabla_x p_t(x)$, which the score network $s_\theta$ learns, depends on the time $t$, so this $t$-dependence must be incorporated into the model via *time conditioning*. When class-labeled training data is available, class-conditional sampling requires *class conditioning* of the score network (Ho & Salimans, 2021). To take advantage of data augmentation and thereby avoid overfitting, EDM (Karras et al., 2022) utilizes *augmentation conditioning* (Jun et al., 2020), where the model is conditioned on the data augmentation information such as the degree of image rotation or blurring. Similarly, SDXL (Podell et al., 2024) uses *micro-conditioning*, where the network is conditioned on image resolution or cropping information. Finally, text-to-image diffusion models (Saharia et al., 2022; Ramesh et al., 2022; Rombach et al., 2022; Podell et al., 2024) use *text conditioning*, which conditions the score network with caption embeddings so that the model generates images aligned with the text description.

**Conditioning attention layers.** Prior diffusion models using CNN-based U-Net architectures condition only convolutional layers in the residual blocks by applying scale-and-shift or adaLN (see **(left)** of Figure 2). In particular, attention blocks are not directly conditioned in such models. This includes the state-of-the-art diffusion models such as Imagen (Saharia et al., 2022), DALL·E 2 (Ramesh et al., 2022), Stable Diffusion (Rombach et al., 2022), and SDXL (Podell et al., 2024). To clarify, Latent Diffusion Model (Rombach et al., 2022) based models use cross-attention method for class and text conditioning, but they still utilize scale-and-shift for time conditioning.

There is a line of research proposing transformer-based architectures (without convolutions) for diffusion models, and these work do propose methods for conditioning attention layers. For instance, DiT (Peebles & Xie, 2023) conditioned attention layers using adaLN and DiffiT (Hatamizadeh et al., 2023) introduced time-dependent multi-head self-attention (TMSA), which can be viewed as scale-and-shift conditioning applied to attention layers. Although such transformer-based architectures have shown to be effective, whether conditioning the attention layers with adaLN or scale-and-shift is optimal was not investigated. In Section 5.5 of this work, we compare our proposed LoRA conditioning on attention layers with the prior adaLN conditioning on attention layers, and show that LoRA is the more effective mechanism for conditioning attention layers.

**Diffusion models as multi-task learners.** Multi-task learning (Caruana, 1997) is a framework where a single model is trained on multiple related tasks simultaneously, leveraging shared representations between the tasks. If one views the denoising tasks for different timesteps (or SNR) of diffusion models as related but different tasks, the training of diffusion models can be interpreted as an instance of the multi-task learning. Following the use of trainable lightweight adapters for Mixture-of-Expert (MoE) (Jacobs et al., 1991; Ma et al., 2018), several works have utilized LoRA as the expert adapter for the multi-task learning (Caccia et al., 2023; Wang et al., 2023a; 2024; Zadouri et al., 2024). Similarly, MORRIS (Audibert et al., 2023) and LoRAHub (Huang et al., 2023) proposed using the weighted sum of multiple LoRA adapters to effectively tackle general tasks. In this work, we took inspiration from theses works by using a composition of LoRA adapters to condition diffusion models.

## 3 Discrete-time LoRA conditioning

Diffusion models such as DDPM (Ho et al., 2020) and IDDPM (Nichol & Dhariwal, 2021) have a predetermined number of discrete timesteps $t = 1, 2, \ldots, T$ used for both training and sampling. We refer to this setting as the *discrete-time* setting.

We first propose a method to condition the attention layers with LoRA in the discrete-time setting. In particular, we implement LoRA conditioning on IDDPM by conditioning the score network with (discrete) time and (discrete) class information.

### 3.1 TimeLoRA

*TimeLoRA* conditions the score network for the discrete time steps $t = 1, \ldots, T$. In prior architectures, time information is typically injected into only the residual blocks containing convolutional layers. TimeLoRA instead conditions the attention blocks. See **(right)** of Figure 2.

**Non-compositional LoRA.** Non-compositional LoRA instantiates $T$ independent rank-$r$ LoRA weights

$$A_1, A_2, \ldots, A_T, \quad B_1, B_2, \ldots, B_T.$$

The dense layer at time $t$ becomes

$$W_t = W + \Delta W(t) = W + B_t A_t$$

for $t = 1, \ldots, T$. To clarify, the trainable parameters for each linear layer are $W$, $A_1, A_2, \ldots, A_T$, and $B_1, B_2, \ldots, B_T$. In particular, $W$ is trained concurrently with $A_1, A_2, \ldots, A_T$, and $B_1, B_2, \ldots, B_T$.

However, this approach has two drawbacks. First, since $T$ is typically large (up to 4000), instantiating $T$ independent LoRAs can occupy significant memory. Second, since each LoRA $(A_t, B_t)$ is trained independently, it disregards the fact that LoRAs of nearby time steps should likely be correlated/similar. It would be preferable for the architecture to incorporate the inductive bias that the behavior at nearby timesteps are similar.

**Compositional LoRA.** Compositional LoRA composes $m$ *LoRA bases*, $A_1, \ldots, A_m$ and $B_1, \ldots, B_m$, where $m \ll T$. Each LoRA basis $(A_i, B_i)$ corresponds to time $t_i$ for $1 \le t_1 < \cdots < t_m \le T$. The dense layer at time $t$ becomes

$$W_t = W + \Delta W(t) = W + \sum_{i=1}^{m} (\omega_t)_i B_i A_i,$$

where $\omega_t = ((\omega_t)_1, \ldots, (\omega_t)_m)$ is the time-dependent trainable weights composing the LoRA bases. To clarify, the trainable parameters for each linear layer are $W$, $A_1, A_1, \ldots, A_m$, $B_1, B_1, \ldots, B_m$, and $\omega_t$.

Since the score network is a continuous function of $t$, we expect $\omega_t \approx \omega_{t'}$ if $t \approx t'$. Therefore, to exploit the task similarity between nearby timesteps, we initialize $(\omega_t)_i$ with a linear interpolation scheme: for $t_j \le t < t_{j+1}$,

$$(\omega_t)_i = \begin{cases} \dfrac{t_{j+1} - t}{t_{j+1} - t_j} & i = j \\ \dfrac{t - t_j}{t_{j+1} - t_j} & i = j + 1 \\ 0 & \text{otherwise.} \end{cases}$$

In short, at initialization, $\Delta W(t)$ uses a linear combination of the two closest LoRA bases. During training, $\omega_t$ can learn to utilize more than two LoRA bases, i.e., $\omega_t$ can learn to have more than two non-zeros through training. Specifically, $(\omega_1, \ldots, \omega_T) \in \mathbb{R}^{m \times T}$ is represented as an $m \times T$ trainable table implemented as `nn.Embedding` in Pytorch.

### 3.2 ClassLoRA

Consider a conditional diffusion model with $C$ classes. *ClassLoRA* conditions the attention layers in the score network with the class label. Again, this contrasts with the typical approach of injecting class information only into the residual blocks containing convolutional layers. See **(right)** of Figure 2.

Since $C$ is small for CIFAR-10 ($C = 10$) and the correlations between different classes are likely not strong, we only use the non-compositional ClassLoRA:

$$W_c = W + \Delta W(c) = W + B'_c A'_c$$

for $c = 1, \ldots, C$. In other words, each LoRA $(A'_c, B'_c)$ handles a single class $c$. When $C$ is large, such as in the case of ImageNet1k, one may consider using a compositional version of ClassLoRA (See Appendix B).

# 4 Continuous-SNR LoRA conditioning

Motivated by (Kingma et al., 2021), some recent models such as EDM (Karras et al., 2022) consider parameterizing the score function as a function of noise or signal-to-noise ratio (SNR) level instead of time. In particular, EDM (Karras et al., 2022) considers the probability flow ODE

$$X_t = -\dot{\sigma}(t)\sigma(t)s_\theta(x;\sigma(t))\ dt,$$

where $s_\theta(x;\sigma)$ is the score network conditioned on the SNR level $\sigma$. We refer to this setting as the *continuous-SNR* setting.

The main distinction between Sections 3 and 4 is in the discrete vs. continuous parameterization, since continuous-time and continuous-SNR parameterizations of score functions are equivalent. We choose to consider continuous-SNR (instead of continuous-time) parameterizations for the sake of consistency with the EDM model (Karras et al., 2022).

Two additional issues arise in the present setup compared to the setting of Section 3. First, by considering a continuum of SNR levels, there is no intuitive way to assign a single basis LoRA to a specific noise level. Second, to accommodate additional conditioning elements such as augmentations or even captions, allocating independent LoRA for each conditioning element could lead to memory inefficiency.

## 4.1 Unified compositional LoRA (UC-LoRA)

Consider the general setting where the diffusion model is conditioned with $N$ attributes $\text{cond}_1, \ldots, \text{cond}_N$, which can be a mixture of continuous and discrete information. In our EDM experiments, we condition the score network with $N = 3$ attributes: SNR level (time), class, and augmentation information.

*Unified compositional LoRA* (UC-LoRA) composes $m$ LoRA bases $A_1, \ldots, A_m$ and $B_1, \ldots, B_m$ to simultaneously condition the information of $\text{cond}_1, \ldots \text{cond}_N$ into the attention layer. The compositional weight $\omega = (\omega_1, \ldots, \omega_m)$ of the UC-LoRA is obtained by passing $\text{cond}_1, \ldots \text{cond}_N$ through an MLP.

Prior diffusion models typically process $\text{cond}_1, \ldots, \text{cond}_N$ with an MLP to obtain a condition embedding $v$, which is then shared by all residual blocks for conditioning. For the $j$-th residual block, $v$ is further processed by an MLP to get scale and shift parameters $\gamma_j$ and $\beta_j$:

$$v = \text{SharedMLP}(\text{cond}_1, \ldots, \text{cond}_N)$$
$$(\gamma_j, \beta_j) = \text{MLP}_j(v).$$

The $(\gamma_j, \beta_j)$ is then used for the scale-and-shift conditioning of the $j$-th residual block in the prior architectures.

In our UC-LoRA, we similarly use the shared embedding $v$ and an individual MLP for the $j$-th attention block to obtain the composition weight $\omega_j(v)$:

$$v = \text{SharedMLP}(\text{cond}_1, \cdots, \text{cond}_N)$$
$$\omega_j(v) = \text{MLP}_j(v).$$

Then, the $j$-th dense layer of the attention block becomes

$$W(\text{cond}_1, \ldots, \text{cond}_N) = W + \Delta W(\text{cond}_1, \ldots, \text{cond}_N)$$
$$= W + \sum_{i=1}^{m} \omega_{j,i}(v)B_iA_i.$$

To clarify, the trainable parameters for the $j$-th dense layer are $W$, $A_1, A_2, \ldots, A_m$, $B_1, B_2, \ldots, B_m$, and the weights in $\text{MLP}_j$. Shared across the entire architecture, the weights in SharedMLP are also trainable parameters.

# 5 Experiments

In this section, we present our experimental findings. Section 5.1 describes the experimental setup. Section 5.2 first presents a toy, proof-of-concept experiment to validate the proposed LoRA conditioning. Section 5.3 evaluates the effectiveness of LoRA conditioning on attention layers with a quantitative comparison between diffusion models with (baseline) conventional scale-and-shift conditioning on convolutional layers; (only LoRA) LoRA conditioning on attention layers without conditioning convolutional layers; and (with LoRA) conditioning both convolutional layers and attention layers with scale-and-shift and LoRA conditioning, respectively. Section 5.4 investigates the effect of tuning the LoRA rank and the number of LoRA bases. Section 5.5 compares our proposed LoRA conditioning with the adaLN conditioning on attention layers. Section 5.6 explores the robustness of ClassLoRA conditioning compared to conventional scale-and-shift conditioning in extrapolating conditioning information.

## 5.1 Experimental Setup

**Diffusion models.** We implement LoRA conditioning on three different diffusion models: nano diffusion (Lelarge et al., 2024), IDDPM (Nichol & Dhariwal, 2021), and EDM-vp (Karras et al., 2022). With nano diffusion, we conduct a proof-of-concept experiment. With IDDPM, we test TimeLoRA and ClassLoRA for the discrete-time setting, and with EDM, we test UC-LoRA for the continuous-SNR setting.

**Datasets.** For nano diffusion, we use MNIST. For IDDPM, we use CIFAR-10 for both unconditional and class-conditional sampling, and ImageNet64, a downsampled version of the ImageNet1k, for unconditional sampling. For EDM-vp, we also use CIFAR-10 for both unconditional and class-conditional sampling and FFHQ64 for unconditional sampling.

**Configurations.** We follow the training and architecture configurations proposed by the baseline works and only tune the LoRA adapters. For IDDPM, we train the model for 500K iterations for CIFAR-10 with batch size of 128 and learning rate of $1 \times 10^{-4}$, and 1.5M iterations for ImageNet64 with batch size of 128 and learning rate of $1 \times 10^{-4}$. For EDM, we train the model with batch size of 512 and learning rate of $1 \times 10^{-3}$ for CIFAR-10, and with batch size of 256 and learning rate of $2 \times 10^{-4}$ for FFHQ64. For sampling, in IDDPM, we use 4000 and 4001 timesteps for the baseline and LoRA conditioning respectively, and in EDM, we use the proposed Heun's method and sample images with 18 timesteps (35 NFE) for CIFAR-10 and 40 timesteps (79 NFE) for FFHQ64. Here, NFE is the number of forward evaluation of the score network and it differs from the number of timesteps by a factor of 2 because Heun's method is a 2-stage Runge–Kutta method. Appendix A provides further details of the experiment configurations.

Note that the baseline works heavily optimized the hyperparameters such as learning rate, dropout probability, and augmentations. Although we do not modify any configurations of the baseline and simply add LoRA conditioning in a drop-in fashion, we expect further improvements from further optimizing the configuration for the entire architecture and training procedure.

**LoRA.** We use the standard LoRA initialization as in the original LoRA paper (Hu et al., 2022): for the LoRA matrices $(A, B)$ with rank $r$, $A$ is initialized as $A_{ij} \sim \mathcal{N}(0, 1/r)$ and $B$ as the zero matrix. Following Ryu (2023), we set the rank of each basis LoRA to 4. For TimeLoRA and ClassLoRA, we use 11 and 10 LoRA bases, and for UC-LoRA we use 18 and 20 LoRA bases for CIFAR-10 and FFHQ.

Due to our constrained computational budget, we were not able to conduct a full investigation on the optimal LoRA rank or the number LoRA bases. However, we experiment with the effect of rank and number of LoRA bases to limited extent and report the result in Section 5.4.

## 5.2 Proof-of-concept experiments

We conduct toy experiments with nano diffusion for both discrete-time and continuous-SNR settings. Nano diffusion is a small diffusion model with a CNN-based U-Net architecture with no skip connections with about $500,000$ trainable parameters. We train nano diffusion on unconditional MNIST generation with

3 different conditioning methods: conventional scale-and-shift, TimeLoRA, and UC-LoRA. As shown in Figure 3, conditioning with TimeLoRA or UC-LoRA yields competitive result compared to the conventional scale-and-shift conditioning.

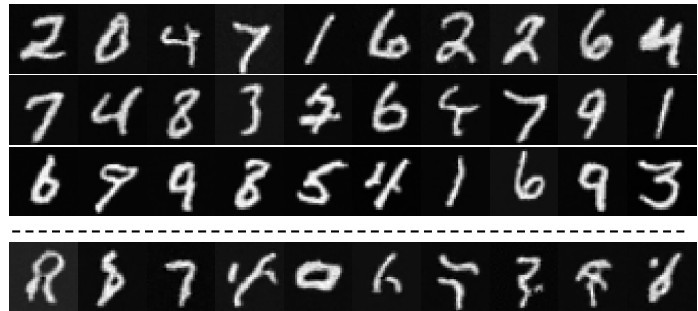

Figure 3: MNIST samples generated by nano diffusion trained with **(1st row)** conventional scale-and-shift conditioning; **(2nd row)** TimeLoRA with linear interpolation initialization; **(3rd row)** UC-LoRA; and **(4th row)** TimeLoRA with random initialization.

**Initialization of $\omega_i(t)$ for TimeLoRA.** As shown in Figure 3 the choice of initialization of $\omega_i(t)$ for TimeLoRA impacts performance. With randomly initialized $\omega_i(t)$, nano diffusion did not converge after 100 epochs, whereas with $\omega_i(t)$ initialized with the linear interpolation scheme, it did converge. Moreover, Figure 4 shows that even in UC-LoRA, $\omega(t)$ shows higher similarity between nearby timesteps than between distant timesteps after training. This is consistent with our expectation that $\omega_i(t) \approx \omega_i(t')$ if $t \approx t'$.

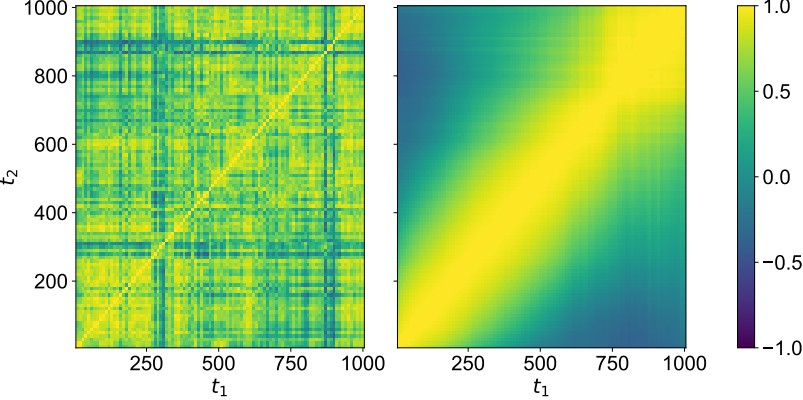

Figure 4: Cosine similarity between $\omega(t_1)$ and $\omega(t_2)$ for UC-LoRA applied to nano diffusion **(left)** at initialization and **(right)** after training. At initialization, the cosine similarity between $\omega(t_1)$ and $\omega(t_2)$ has no discernible pattern. After training, however, the cosine similarity between $\omega(t_1)$ and $\omega(t_2)$ for $t_1 \approx t_2$ is close to 1, implying their high similarity.

## 5.3 Main quantitative results

**Simply adding LoRA conditioning yields improvements.** To evaluate the effectiveness of the drop-in addition of LoRA conditioning to the attention layers, we implement TimeLoRA and ClassLoRA to IDDPM and UC-LoRA to EDM, both with the conventional scale-and-shift conditioning on the convolutional layers unchanged. We train IDDPM with CIFAR-10, ImageNet64 and EDM with CIFAR-10, FFHQ64. As reported in Table 1, the addition of LoRA conditioning to the attention layers consistently improves the image generation quality as measured by FID scores (Heusel et al., 2017) across different diffusion models and datasets with only (~10%) addition of the parameter counts. Note these improvements are achieved without tuning any hyperparameters of the base model components.

**Initializing the base model with pre-trained weights.** We further test UC-LoRA on pre-trained EDM base models for unconditional CIFAR-10 and FFHQ64 generations. As reported in Table 1, using pre-trained weights showed additional gain on FID score with fewer number of interations ($\sim 50\%$). To clarify, although we initialize the base model with pre-trained weights, we fully train both base model and LoRA modules rather than finetuning.

**LoRA can even replace scale-and-shift.** We further evaluate the effectiveness of LoRA conditioning by replacing the scale-and-shift conditioning for the convolutional layers in residual blocks with LoRA conditioning for the attention blocks. The results of Table 1 suggest that solely using LoRA conditioning on attention layers achieves competitive FID scores while being more efficient in memory compared to the baseline score network trained with scale-and-shift conditioning on convolutional layers. For IDDPM, using LoRA in place of the conventional scale-and-shift conditioning consistently produces better results. Significant improvement is observed especially for class-conditional generation of CIFAR-10. For EDM, replacing the scale-and-shift conditioning did not yield an improvement, but nevertheless performed comparably. We note that in all cases, LoRA conditioning is more parameter-efficient ($\sim 10\%$) than the conventional scale-and-shift conditioning.

### 5.4 Effect of LoRA rank and number of LoRA bases

We investigate the effect of tuning the LoRA rank and the number of LoRA bases on the EDM model for unconditional CIFAR-10 generation and report the results in Table 2. Our findings indicate that using more LoRA bases consistently improves the quality of image generations. On the other hand, increasing LoRA rank does not guarantee better performance. These findings suggest an avenue of further optimizing and improving our main quantitative results of Section 5.3 and Table 1, which we have not yet been able to pursue due to our constrained computational budget.

|  | # basis | rank | FID | # Params |
|---|---|---|---|---|
| Varying # basis | 9 | 4 | 1.99 | 57185519 |
|  | 18 | 4 | 1.96 | 57745499 |
|  | 36 | 4 | **1.95** | 58865459 |
| Varying rank | 18 | 2 | **1.93** | 57192539 |
|  | 18 | 4 | 1.96 | 57745499 |
|  | 18 | 8 | 1.96 | 58851419 |

Table 2: Effect of the number of LoRA bases and the LoRA rank on unconditional CIFAR-10 sampling of EDM with LoRA

### 5.5 Comparison with adaLN

We compare the effectiveness of our proposed LoRA conditioning with adaLN conditioning applied to attention layers. Specifically, we conduct an experiment on EDM with scale-and-shift conditioning on convolutional layers removed and with (i) adaLN conditioning attention layers or (ii) LoRA conditioning attention layers. We compare the sample quality of unconditional and class-conditional CIFAR-10 generation and report the results in Table 3. We find that LoRA conditioning significantly outperforms adaLN conditioning for both unconditional and conditional CIFAR-10 generation. This indicates that our proposed LoRA conditioning is the more effective mechanism for conditioning attention layers in the U-Net architectures for diffusion models.

| Type | uncond. | cond. |
|---|---|---|
| adaLN conditioning | 2.16 | 2.0 |
| LoRA conditioning | **1.99** | **1.82** |

Table 3: Comparison of adaLN conditioning and LoRA conditioning on attention layers on EDM (without conditioning convolutional layers). We consider both unconditional and conditional CIFAR-10 generation.

### 5.6 Extrapolating conditioning information

We conduct an experiment comparing two class-conditional EDM models each conditioned by scale-and-shift and ClassLoRA, for the CIFAR-10 dataset. During training, both models receive size-10 one-hot vectors $(c_i)_j = \delta_{ij}$ representing the class information.

First, we input the linear interpolation $\alpha c_i + (1-\alpha)c_j$ $(0 \le \alpha \le 1)$ of two class inputs $c_i$ and $c_j$ (corresponding to 'airplane' and 'horse', respectively) to observe the continuous transition between classes. As shown in the top of Figure 5, both the scale-and-shift EDM and ClassLoRA EDM models effectively interpolate semantic information across different classes. However, when a scaled input $\beta c_i$ is received, with $\beta$ ranging from -1 to 1, scale-and-shift EDM generates unrecognizable images when $\beta < 0$, while ClassLoRA EDM generates plausible images throughout the whole range, as shown in the bottom of Figure 5. This toy experiment shows that LoRA-based conditioning may be more robust to extrapolating conditioning information beyond the range encountered during training. Appendix D provides further details.

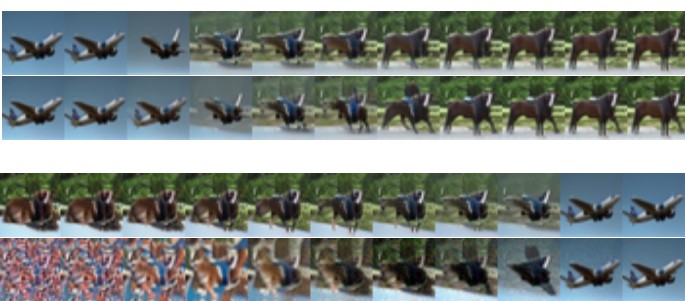

Figure 5: Results of **(Top)** interpolation of class labels in class-conditional EDM with (row1) ClassLoRA; (row2) scale-and-shift; **(bottom)** extrapolation of class labels in class-conditional EDM with (row1) ClassLoRA; (row2) scale-and-shift

## 6 Conclusion

In this work, we show that simply adding Low-Rank Adaptation (LoRA) conditioning to the attention layers in the U-Net architectures improves the performance of the diffusion models. Our work shows that we should condition the attention layers in diffusion models and provides a prescription for effectively doing so. Some prior works have conditioned attention layers in diffusion models with adaLN or scale-and-shift operations, but we find that LoRA conditioning is much more effective as discussed in Section 5.5.

Implementing LoRA conditioning on different and larger diffusion model architectures is a natural and interesting direction of future work. Since almost all state-of-the-art (SOTA) or near-SOTA diffusion models utilize attention layers, LoRA conditioning is broadly and immediately applicable to all such architectures. In particular, incorporating LoRA conditioning into large-scale diffusion models such as Imagen (Saharia et al., 2022), DALL·E 2 (Ramesh et al., 2022), Stable Diffusion (Rombach et al., 2022), and SDXL (Podell et al., 2024), or transformer-based diffusion models such as U-ViT (Bao et al., 2023), DiT (Peebles & Xie, 2023), and DiffiT (Hatamizadeh et al., 2023) are interesting directions. Finally, using LoRA for the text conditioning of text-to-image diffusion models is another direction with much potential impact.

## 7 Acknowledgements

This research was supported by a grant from KRAFTON AI. AN was supported in part by the Ministry of Science and ICT (MSIT), South Korea, under the Information Technology Research Center (ITRC) Support Program #IITP-2024-RS-2022-00156295. EKR was supported by the National Research Foundation of Korea(NRF) grant funded by the Korean government (No.RS-2024-00421203). We thank Sehyun Kwon and Dongwhan Rho for their careful reviews and valuable feedback. We also thank the anonymous reviewers for their insightful comments.

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

# A  Experimental details

Here, we provide the detailed experiment settings. Note that apart from the conditioning method, we mostly follow base codes and configurations provided by Dataflowr, IDDPM repository, and EDM repository for nano diffusion, IDDPM and EDM respectively.

## A.1  Nano Diffusion

We mostly followed the base code provided by Dataflowr with 3 exceptions. First, the sinusoidal embedding implemented in the original code was not correctly implemented. Although it did not have visible impact on TimeLoRA and UC-LoRA, it significantly deteriorated the sample quality of the scale-and-shift conditioning (see Figure 6). Second, during training process, the input image is normalized with mean = 0.5 and std = 0.5. However, it is not considered in the visualization process. Lastly, we extended the number of training epochs from 50 to 100 for better convergence.

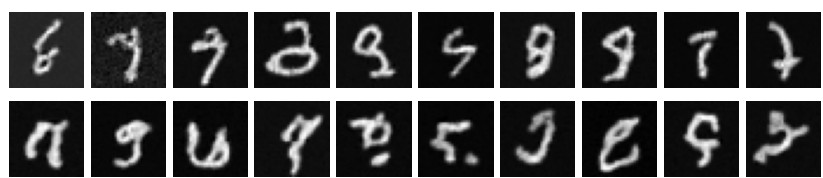

Figure 6: MNIST image generated with incorrect sinusoidal embedding and scale-and-shift conditioning, **(top)** after 100 epochs of training, **(bottom)** after 400 epochs of training.

## A.2  IDDPM

Here we provide the training setting used in IDDPM experiments based on the IDDPM repository.

### A.2.1  CIFAR-10

For CIFAR-10 training, we construct U-Net with model channel of 128 channels, and 3 residual blocks per each U-Net blocks. We use Adam optimizer with learning rate of $1 \times 10^{-4}$, momentum of $(\beta_1, \beta_2) = (0.9, 0.999)$, no weight decay, and dropout rate of 0.3. We train the model with $L_{\text{hybrid}}$ proposed in the original paper for 500k iterations with batch size of 128. The noise is scheduled with the cosine scheduler and the timestep is sampled with uniform sampler at training. For sampling, we use checkpoints saved every 50k iterations with exponential moving average of rate 0.9999, and sample image for 4000 steps and 4001 steps for the baseline and LoRA conditioning, respectively.

**Training flags used for unconditional CIFAR-10 training.**

```
MODEL_FLAGS="--image_size 32 --num_channels 128 --num_res_blocks 3 --learn_sigma True --dropout 0.3"
DIFFUSION_FLAGS="--diffusion_steps 4000 --noise_schedule cosine"
TRAIN_FLAGS="--lr 1e-4 --batch_size 128"
```

**Training flags used for class-conditional CIFAR-10 training.**

```
MODEL_FLAGS="--image_size 32 --num_channels 128 --num_res_blocks 3 --learn_sigma True --dropout 0.3 --class_cond True"
DIFFUSION_FLAGS="--diffusion_steps 4000 --noise_schedule cosine"
TRAIN_FLAGS="--lr 1e-4 --batch_size 128"
```

### A.2.2  ImageNet64

For ImageNet64 training, we use the same U-Net construction used in CIFAR-10 experiment. The model channel is set as 128 channels, and each U-Net block contains 3 residual blocks. We use Adam optimizer with learning rate of $1 \times 10^{-4}$, momentum of $(\beta_1, \beta_2) = (0.9, 0.999)$, no weight decay and no dropout. We train the model with $L_{\text{hybrid}}$ proposed in the original paper for 1.5M iterations with batch size of 128. The noise

is scheduled with the cosine scheduler and the timestep is sampled with uniform sampler at training. For sampling, we use checkpoints saved every 500k iterations with exponential moving average of rate 0.9999, and sample images for 4000 steps and 4001 steps for the baseline and LoRA conditioning, respectively.

**Training flags used for ImageNet64 training.**

```
MODEL_FLAGS="--image_size 64 --num_channels 128 --num_res_blocks 3 --learn_sigma True"
DIFFUSION_FLAGS="--diffusion_steps 4000 --noise_schedule cosine"
TRAIN_FLAGS="--lr 1e-4 --batch_size 128"
```

### A.3 EDM

Here we provide the training setting used in EDM experiments based on the EDM repository.

#### A.3.1 CIFAR-10

For CIFAR-10 training, we use the DDPM++ (Song et al., 2021b) with channel per resolution as 2, 2, 2. We use Adam optmizer with learning rate of $1 \times 10^{-3}$, momentum of $(\beta_1, \beta_2) = (0.9, 0.999)$, no weight decay and dropout rate of 0.13. We train the model with EDM preconditioning and EDM loss proposed in the paper for 200M training images (counting repetition) with batch size of 512 and augmentation probability of 0.13. For sampling, we used checkpoints saved every 50 iterations with exponential moving average of 0.99929, and sample image using Heun's method for 18 steps (35 NFE).

**Training flags used for unconditional CIFAR-10 training.**

```
--cond=0 --arch=ddpmpp
```

**Training flags used for class-conditional CIFAR-10 training.**

```
--cond=1 --arch=ddpmpp
```

#### A.3.2 FFHQ64

For FFHQ training, we use the DDPM++ (Song et al., 2021b) with channel per resolution as 1, 2, 2, 2. We use Adam optmizer with learning rate of $2 \times 10^{-4}$, momentum of $(\beta_1, \beta_2) = (0.9, 0.999)$, no weight decay and dropout rate of 0.05. We train the model with EDM preconditioning and EDM loss proposed in the paper for 200M training images (counting repetition) with batch size of 256 and augmentation probability of 0.15. For sampling, we used checkpoints saved every 50 iterations with exponential moving average of about 0.99965, and sample image using Heun's method for 40 steps (79 NFE).

**Training flags used for FFHQ training.**

```
--cond=0 --arch=ddpmpp --batch=256 --cres=1,2,2,2 --lr=2e-4 --dropout=0.05 --augment=0.15
```

### A.4 LoRA basis for TimeLoRA

As IDDPM uniformly samples the timestep during training, we follow the similar procedure for selecting the timestep $t_i$ assigned for the LoRA basis $(A_i, B_i)$ in TimeLoRA. To be specific, we set $t_1 = 1, t_m = T$, and equally distribute $t_i$ in between:

$$t_i = 1 + (i - 1) \cdot \frac{T - 1}{m}.$$

Note here, for simplicity we assumed that $m$ divides $T - 1$, and choose $T = 4001$ instead of $T = 4000$ used in the baseline work.

### A.5 MLP for composition weights of UC-LoRA

For the composition weights of UC-LoRA, we used 3-layer MLP with group normalization and SiLU activation. Specifically, each LoRA module contains a MLP consisting of two linear layers with by group normalization and SiLU activation followed by a output linear layer. MLP takes the shared condition embedding $v \in \mathbb{R}^{d_{\text{emb}}}$ as the input and outputs the composition weight $\omega(v) \in \mathbb{R}^m$, where $d_{\text{emb}}$ is the embedding dimension (512 for EDM) and $m$ is the number of LoRA bases. This is implemented in Pytorch as

```
self.comp_weights = nn.Sequential(
    Linear(in_features=embed_dim, out_features=128),
    GroupNorm(num_channels=N1, eps=1e-6),
    torch.nn.SiLU(),
    Linear(in_features=N1, out_features=N2),
    GroupNorm(num_channels N2, eps=1e-6),
    torch.nn.SiLU(),
    Linear(in_features=N2, out_features=num_basis),
)
```

In our experiment, we set $(N1, N2) = (50, 50)$ for nano diffusion and $(N1, N2) = (128, 64)$ for EDM. Note the choice of depth, and the width of the MLP is somewhat arbitrary and can be further optimzied.

## B Compositional ClassLoRA

To verify the effectiveness of a compositional version of ClassLoRA, we conducted a proof-of-concept experiment with EDM on CIFAR-10. We finetuned a pretrained unconditional EDM for class-conditional CIFAR-10 sampling using a *single* LoRA basis and randomly initialized compositional weights. Note in the non-compositional ClassLoRA, we used 10 LoRA bases, one for each class. As a result, with the addition of only 61560 parameter counts, we were able to convert (fine-tune) an unconditional EDM to a conditional EDM and improve the FID score from 1.97 to 1.81.

## C Cosine similarity between $\omega_t$s in nano diffusion

We present cosine similarity between $\omega_t$ and $\omega_{500}$ for all LoRA bases in UC-LoRA for nano diffusion in Figure 7. As observed in Section 5.2 and Figure 4, cosine similarity is consistently high for $t$ close to 500, proving that there is a task similarity between nearby timesteps for all layers of the network. However, the patterns varied depending on the depth of the layer within the network. This could be an interesting point for future research to further understand the learning dynamics of the diffusion models.

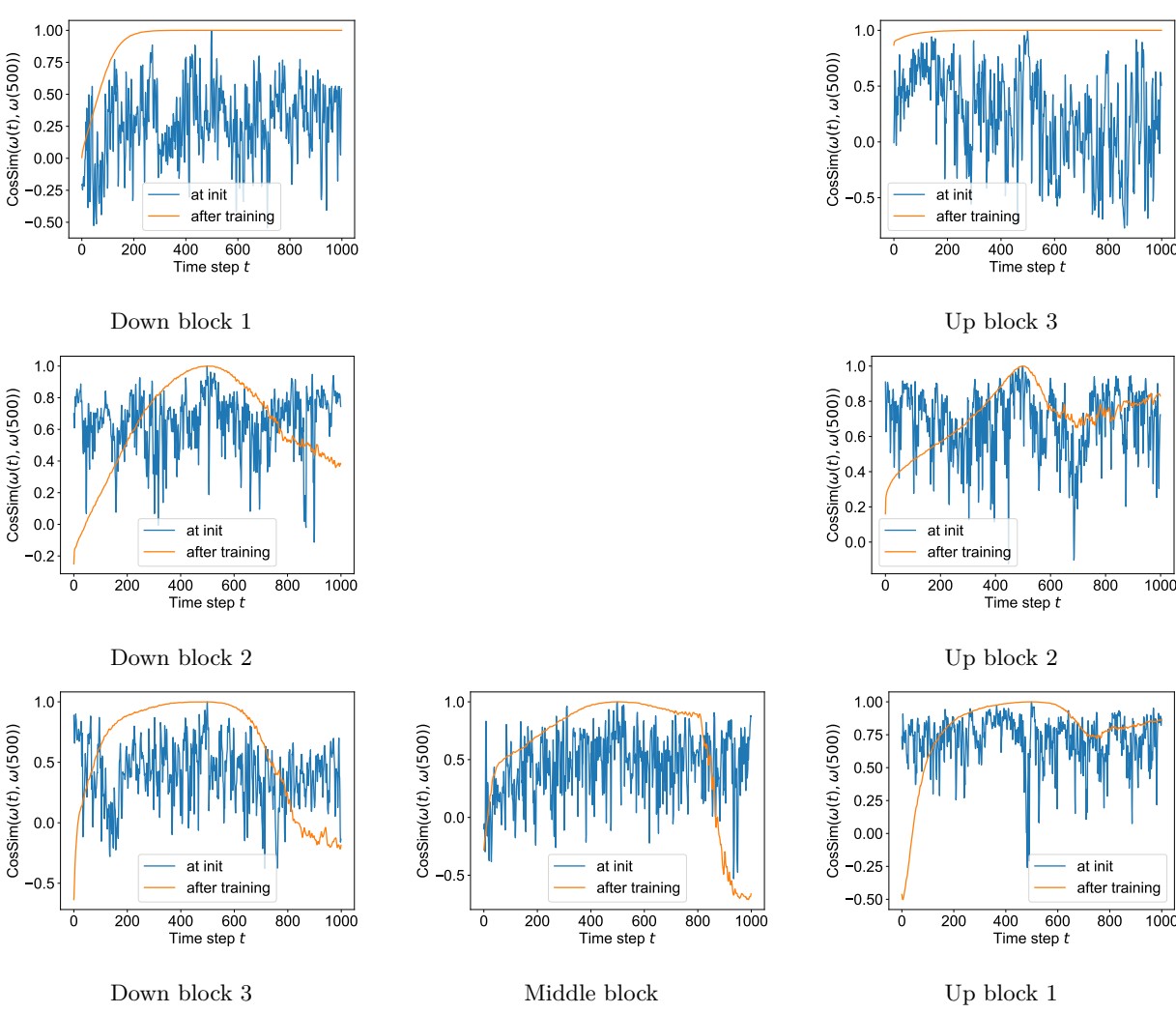

Figure 7: Cosine similarity between $\omega_t$ and $\omega_{500}$ for UC-LoRA applied to nano diffusion.

## D   Extrapolating conditioning information with Class LoRA

We present a more detailed analysis along with comprehensive results of the experiments introduced in Section 5.6. The class-conditional EDM model used for comparison was trained with the default training configurations explained in Appendix A.3. For ClassLoRA-conditioned EDM, we used the unconditional EDM as the base model and applied ClassLoRA introduced in Section 3.2 for the discrete-time setting. We did not use UC-LoRA for this ablation study, with the intent of focusing on the effect of LoRA conditioning on class information.

We provide interpolated and extrapolated images as introduced in 5.6 for various classes in Figure 8 and Figure 9, corroborating the consistent difference between the two class conditioning schemes across classes. We also experiment with strengthening the class input, where we use scaled class information input $\beta c_i$ with $\beta > 1$. Fig. 10 shows that LoRA conditioning shows more robustness in this range as well.

Considering the formulation of ClassLoRA, interpolating or scaling the class input is equivalent to interpolating or scaling the class LoRA adapters, resulting in a formulation similar to Compositional LoRA or UC-LoRA:

$$W_t = W + \Delta W(c) = W + \sum_{i=1}^{C} \omega_i B'_c A'_c,$$

where $\omega_i$ corresponds to the interpolation or scaling of the class inputs. From this perspective, the $\omega$ weights could be interpreted as a natural 'latent vector' capturing the semantic information of the image, in a similar sense that was highlighted in Kwon et al. (2023). While our current focus was exclusively on class information, we hypothesize that this method could be extended to train style or even text conditioning, especially considering the effectiveness of LoRA for fine-tuning.

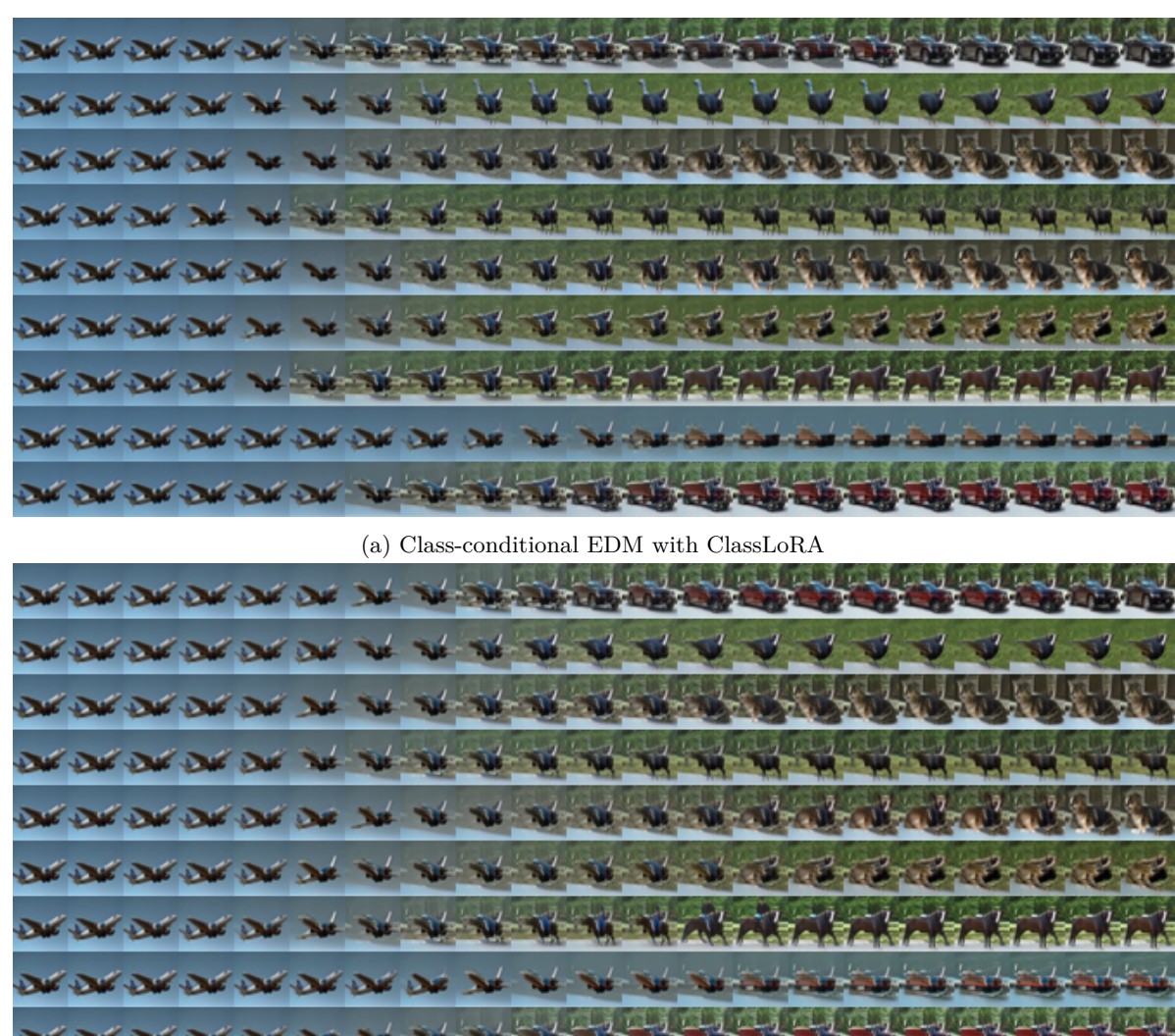

(a) Class-conditional EDM with ClassLoRA

(b) Class-conditional EDM with scale-and-shift

Figure 8: Results of interpolation of class labels for various classes

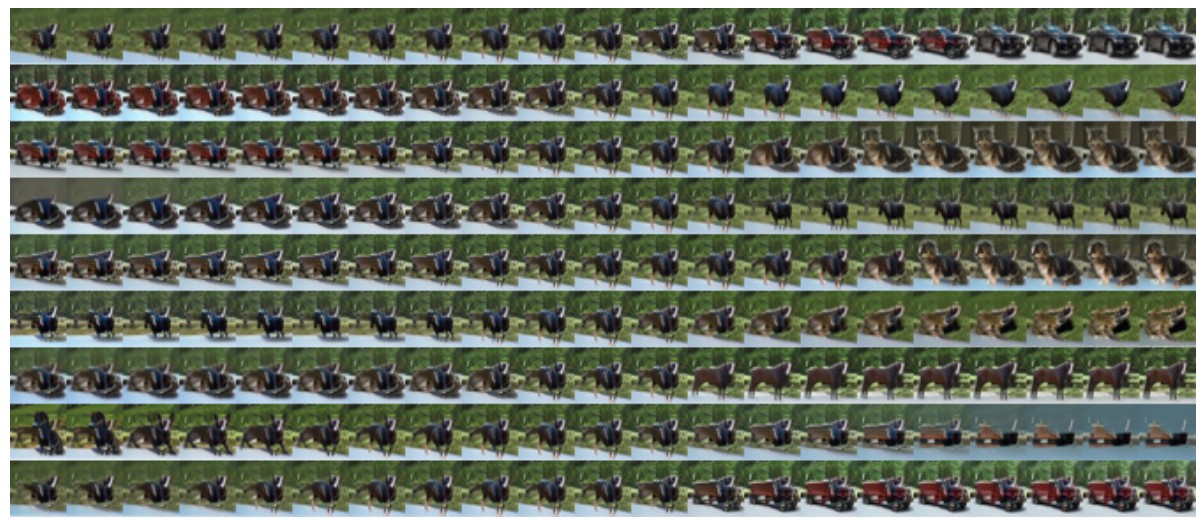

(a) Class-conditional EDM with ClassLoRA

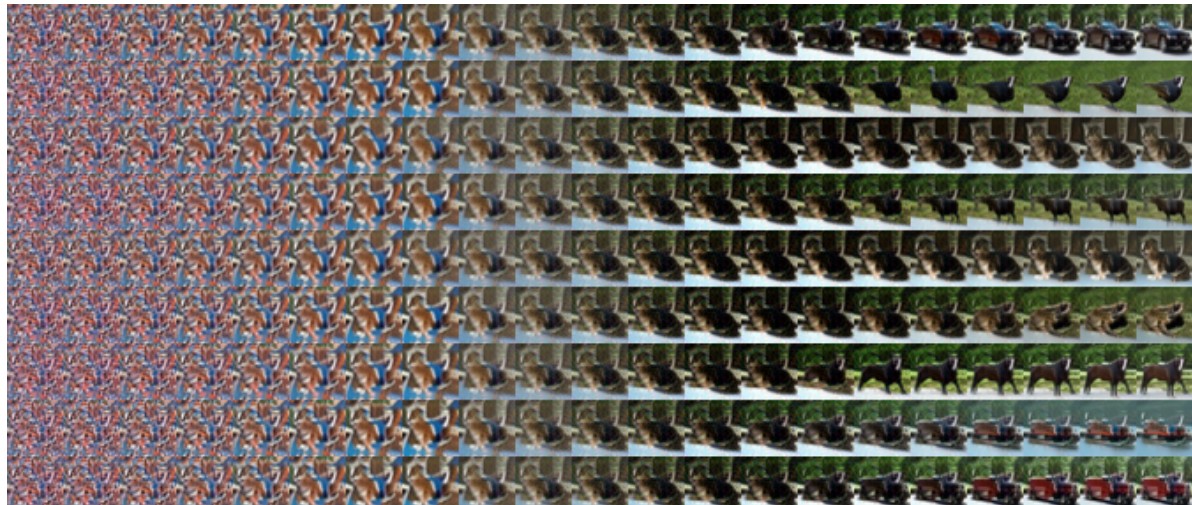

(b) Class-conditional EDM with scale-and-shift

Figure 9: Results of extrapolation of class labels for various classes

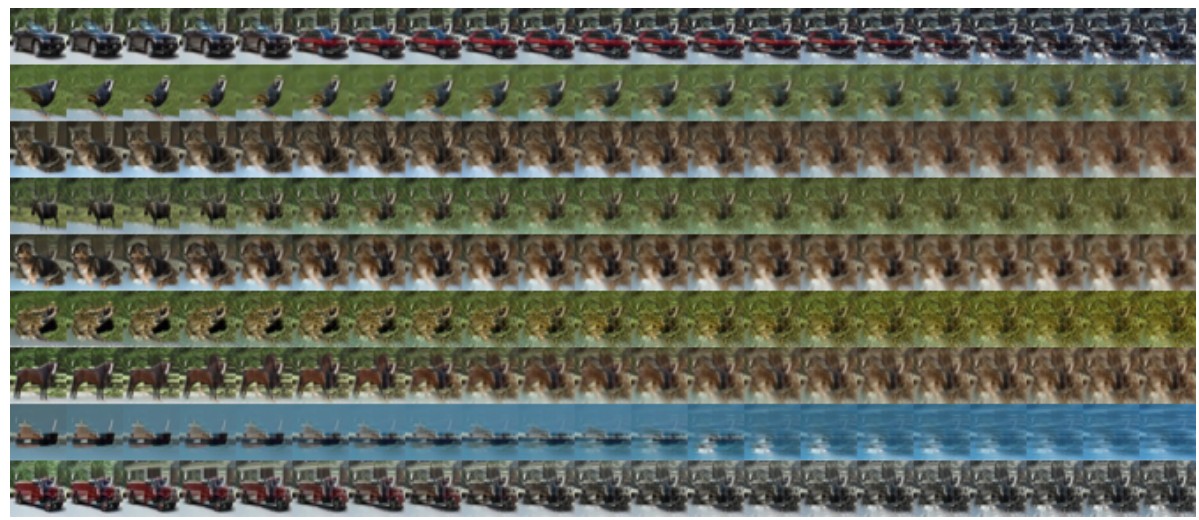

(a) Class-conditional EDM with ClassLoRA

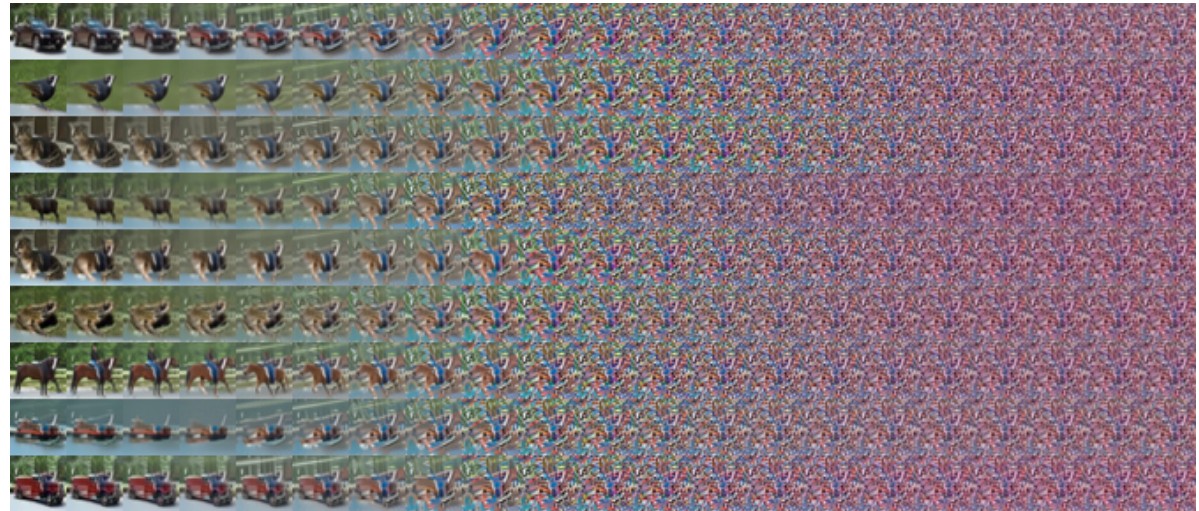

(b) Class-conditional EDM with scale-and-shift

Figure 10: Results of class labels scaled from 1 to 5 for various classes

# E  Image generation samples

## E.1  IDDPM

### E.1.1  Unconditional CIFAR-10

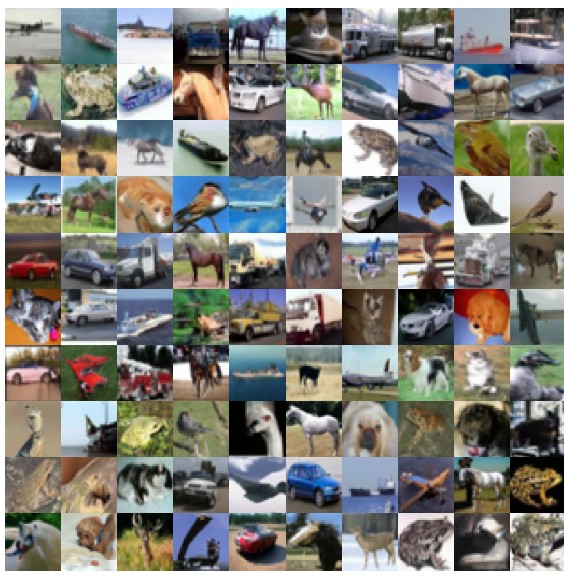

Figure 11: Unconditional CIFAR-10 samples generated by IDDPM with LoRA

### E.1.2  Class-conditional CIFAR-10

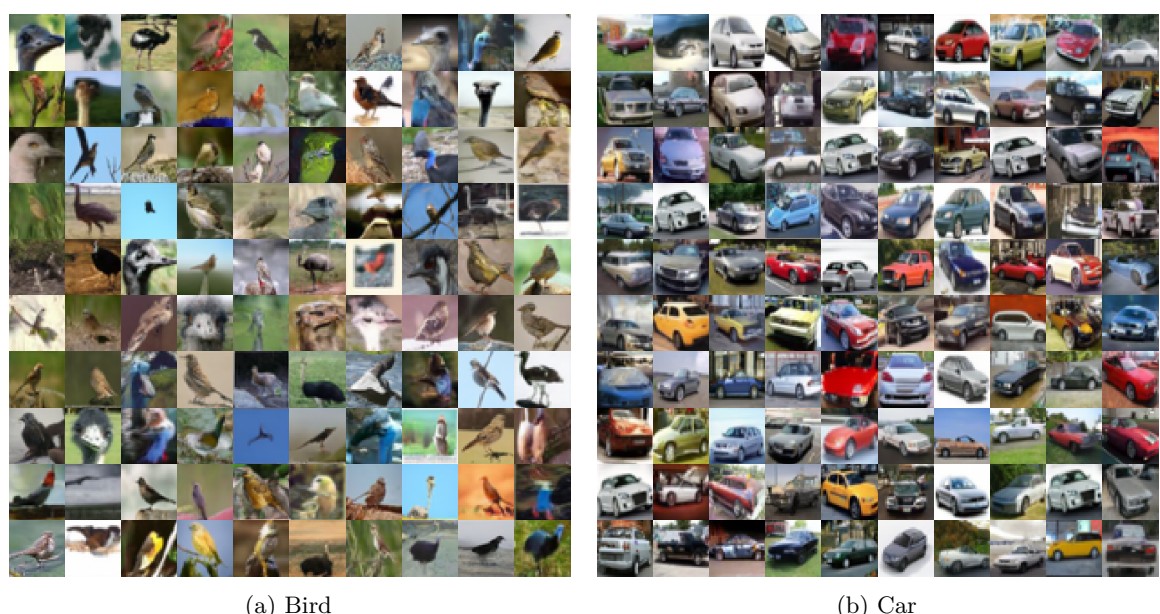

(a) Bird                                                    (b) Car

Figure 12: Class conditional CIFAR-10 samples generated by IDDPM with LoRA

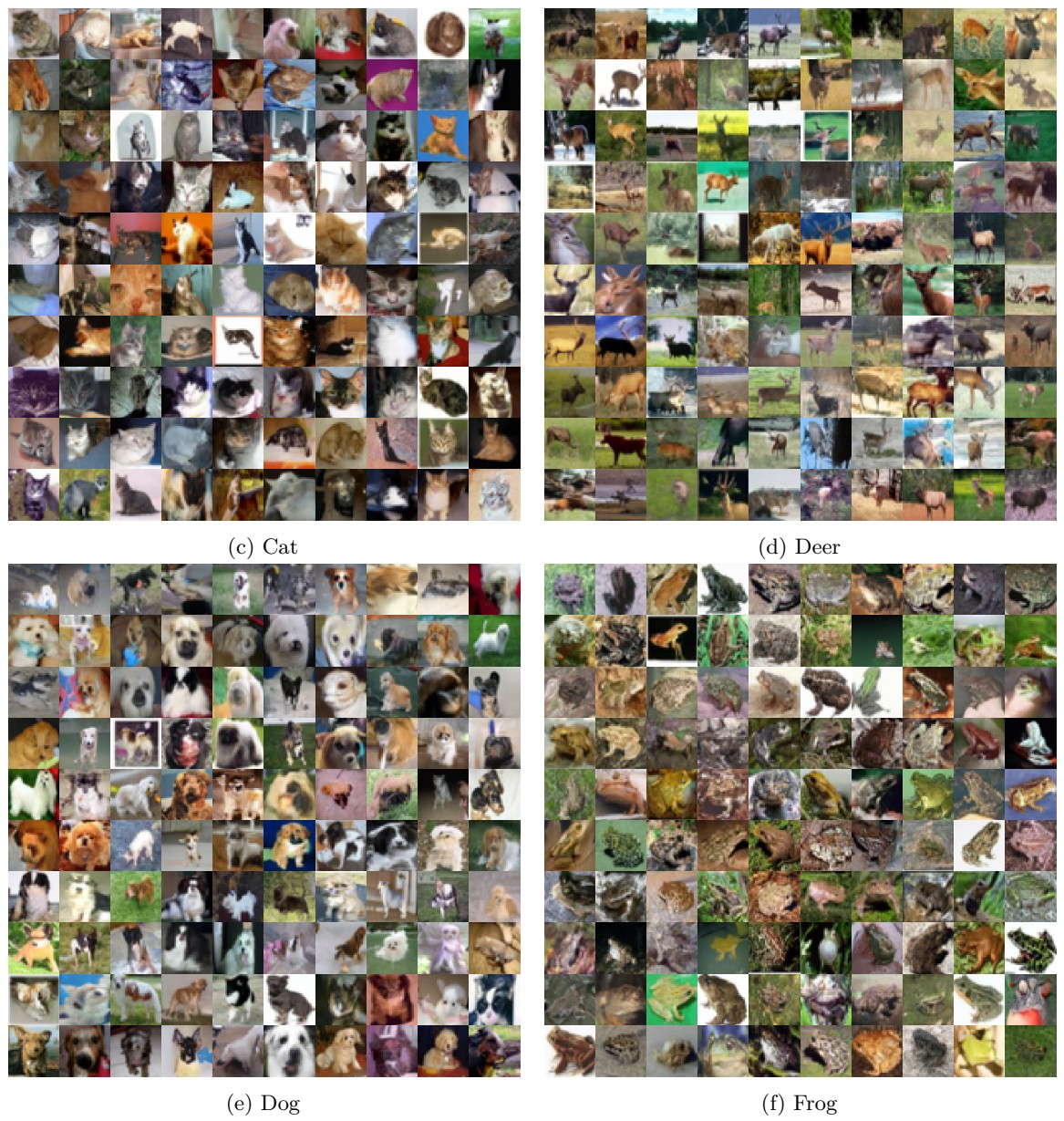

(c) Cat

(d) Deer

(e) Dog

(f) Frog

Figure 12: Class conditional CIFAR-10 samples generated by IDDPM with LoRA

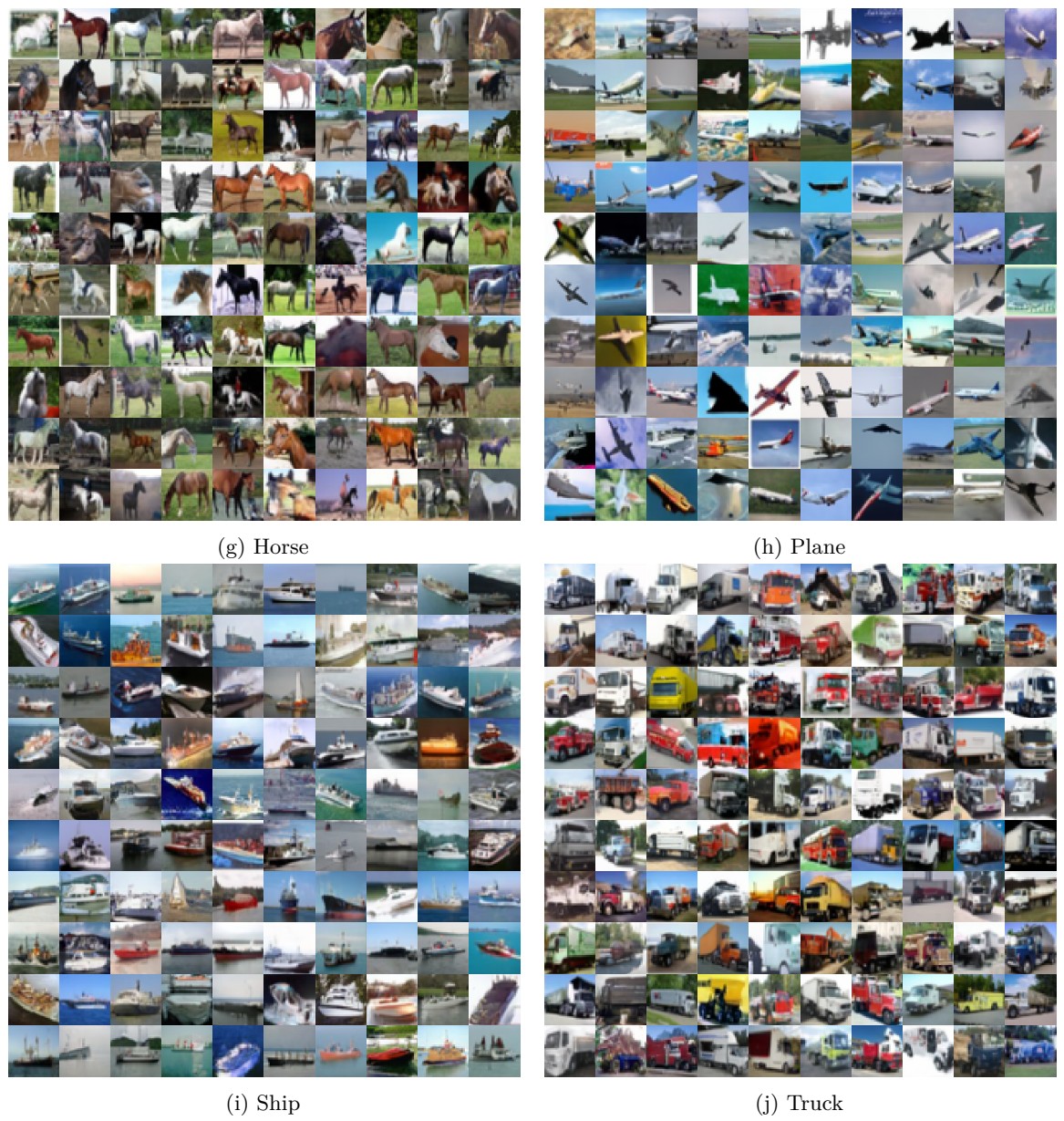

(g) Horse

(h) Plane

(i) Ship

(j) Truck

Figure 12: Class conditional CIFAR-10 samples generated by IDDPM with LoRA

### E.1.3 ImageNet64

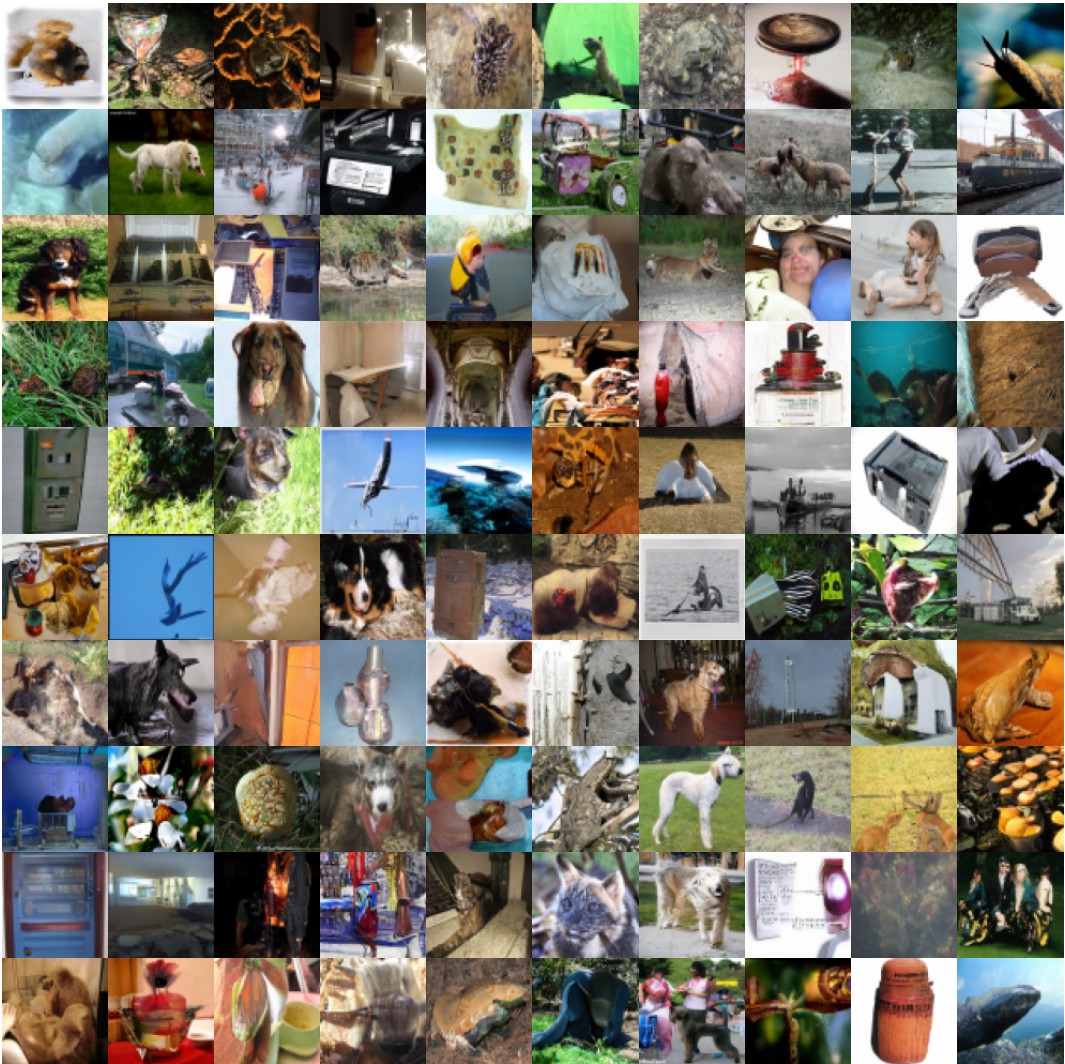

Figure 13: ImageNet64 samples generated by IDDPM with LoRA

## E.2 EDM

### E.2.1 Unconditional CIFAR-10

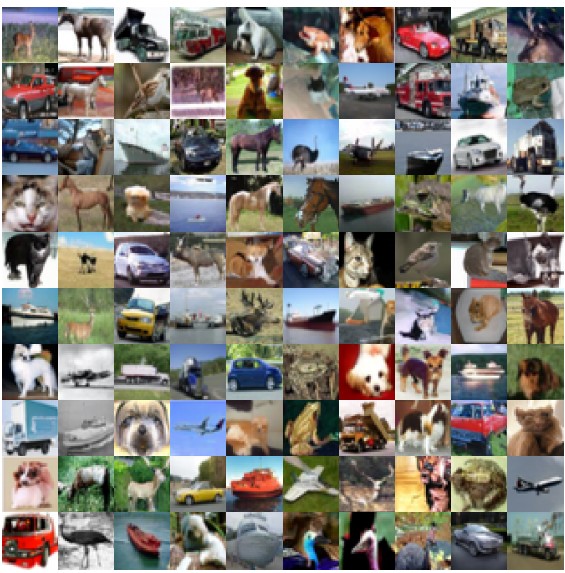

Figure 14: Unconditional CIFAR-10 samples generated by EDM with LoRA

### E.2.2 Class-conditional CIFAR-10

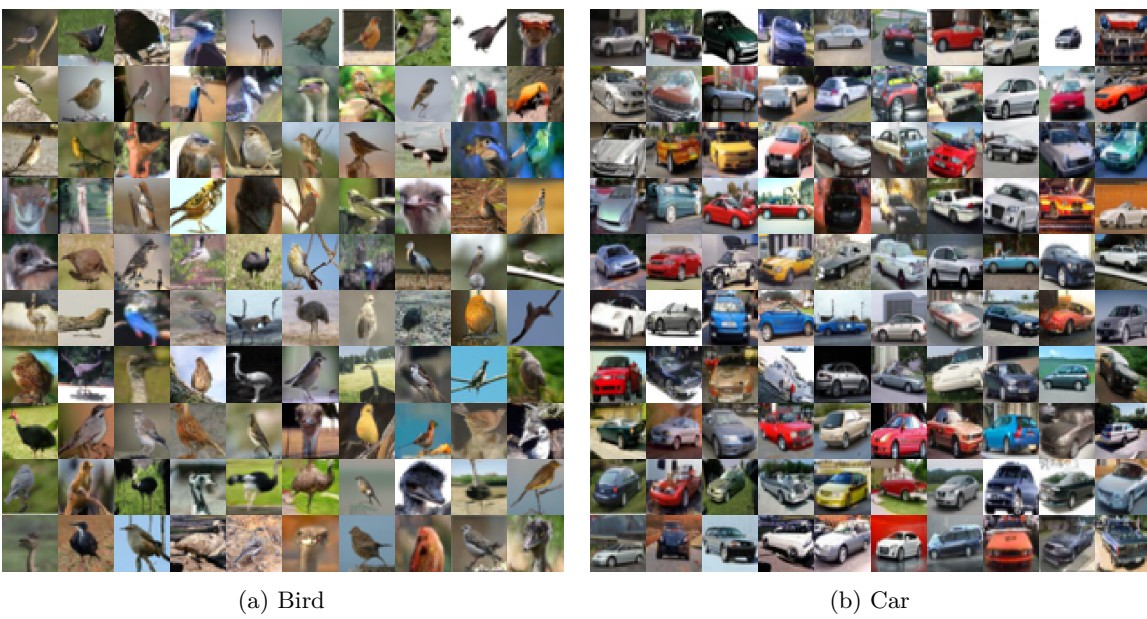

(a) Bird      (b) Car

Figure 15: Class conditional CIFAR-10 samples generated by EDM (vp) with LoRA

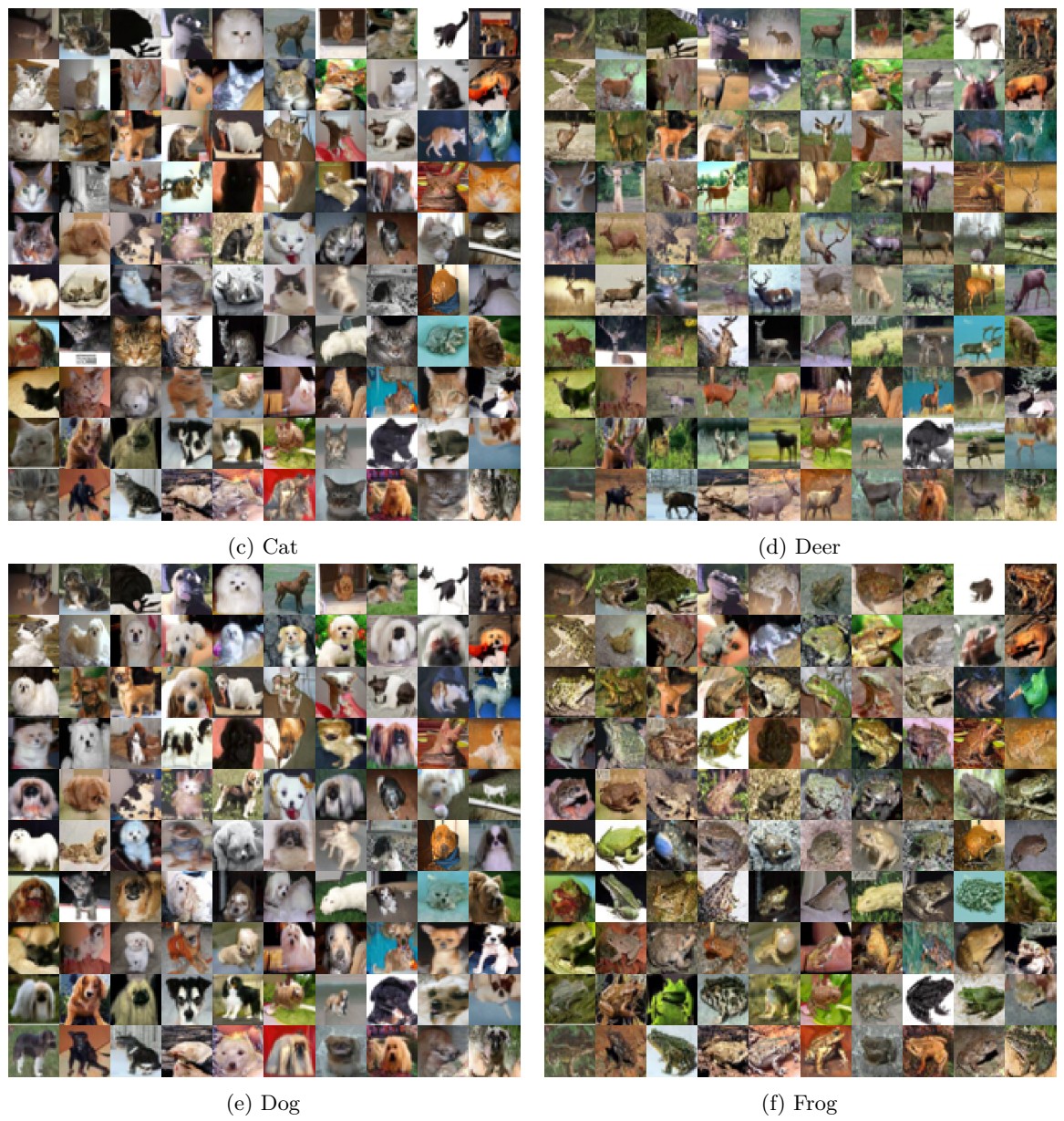

(c) Cat

(d) Deer

(e) Dog

(f) Frog

Figure 15: Class conditional CIFAR-10 samples generated by EDM (vp) with LoRA

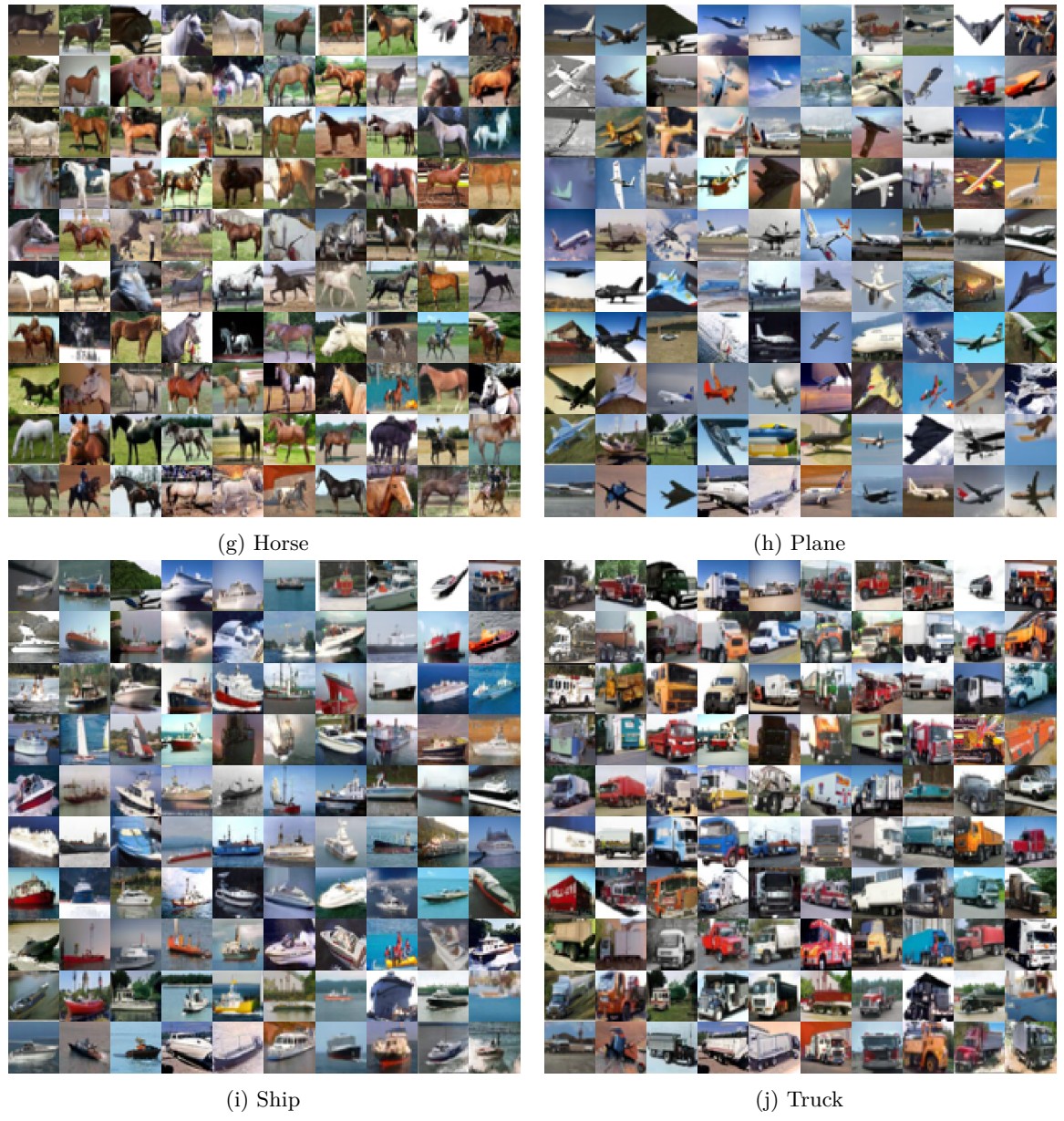

(g) Horse

(h) Plane

(i) Ship

(j) Truck

Figure 15: Class conditional CIFAR-10 samples generated by EDM (vp) with LoRA

### E.2.3 FFHQ

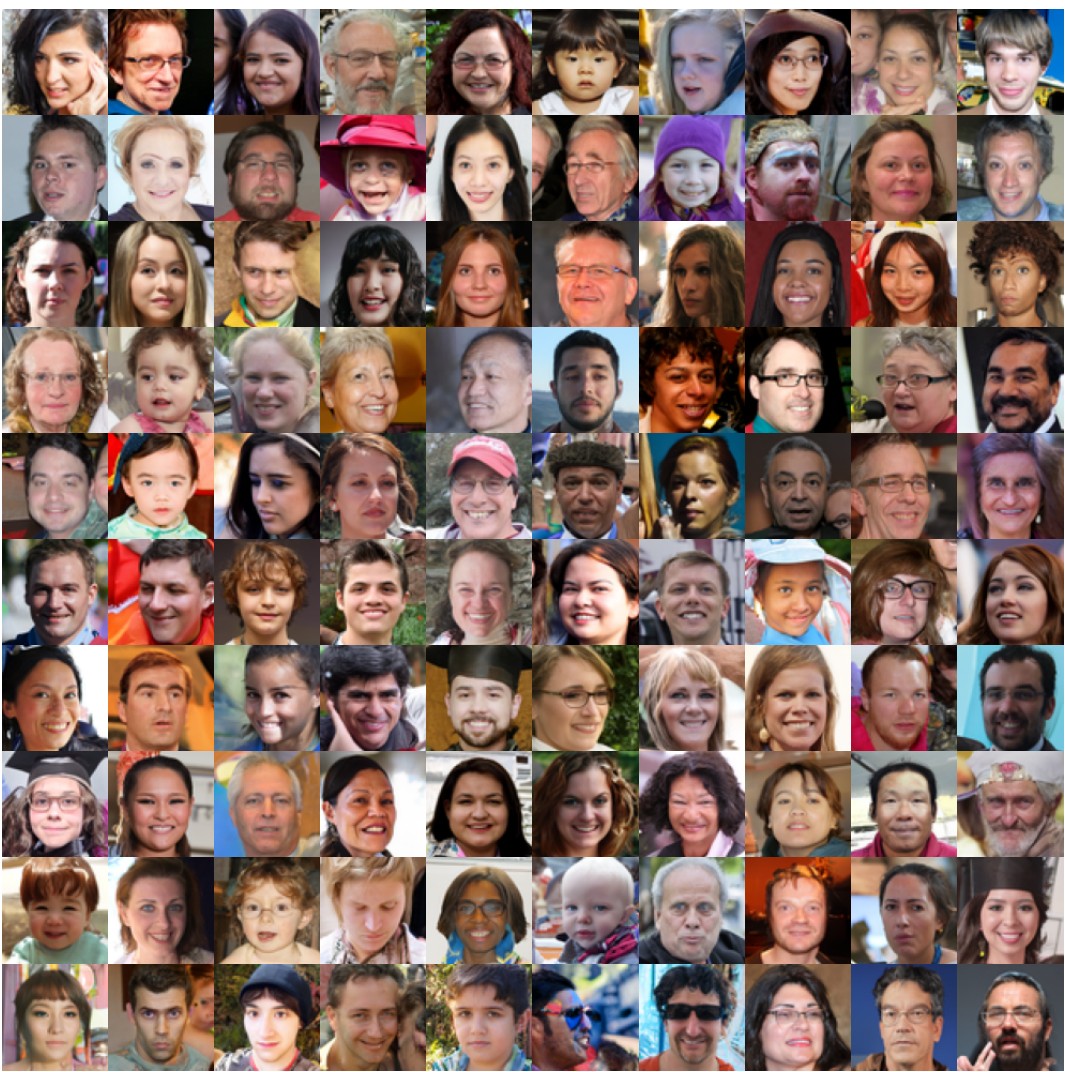

Figure 16: FFHQ samples generated by EDM (vp) with LoRA

