# OpenReview forum: "Simple Drop-in LoRA Conditioning on Attention Layers Will Improve Your Diffusion Model"
_TMLR — Accepted by TMLR_

### Review · Reviewer_XaAm · 2024-06-16

**Summary Of Contributions:**

This work proposes to apply drop-in lora (low rank layer) to the attention layers of the attention layers of a U-net improves the performance of the image generation tasks

**Audience:**

Yes

**Claims And Evidence:**

Yes

**Requested Changes:**

I believe having a more convincing and not-so-arbitrary motivation would be much better

**Strengths And Weaknesses:**

Strength: Using LoRA in this problem is novel and leads to promising result

Weakness: Lack of theory, and the motivation is not too strong. For the present manuscript, the motivation is just (paraphrased) "we should condition the attention layers because we also conditioned other." This is not a strong motivation for conditioning the attention layers, nor a strong motivation to condition the attention in the specific way the authors proposed

---

> ### Author Response · Authors · 2024-07-29
> **Response to Reviewer XaAm**
>
> We thank the reviewer for the constructive feedback. We are happy to hear that the reviewer finds our method novel and promising.
>
> # Theoretical motivation (shared with response to Reviewer Rtei)
>
>
> As we discuss in the introduction, prior conditioning methods only condition the convolutional layers and simply do not condition this attention layer. The omission of the attention layers in the conditioning is an arbitrary design choice, and our proposal to also condition the attention layers with LoRA is, in our view, a natural one.
>
> As the reviewer points out, our work does not present a theoretical motivation behind our proposal. The lack of theoretical justification of the neural network architecture is a shared shortcoming of essentially all prior work in diffusion models. (The only theoretical work that we are aware of is [1], which provides post hoc justification.) Nevertheless, we did have some empirical motivation behind using LoRA for conditioning diffusion models:
>
> - Prior work has shown LoRA to be compatible with diffusion models for the purpose of fine-tuning.
> - The conditioning of the U-Net to learn the time- and class-dependent score function can be viewed as a multi-task learning setup, and prior work has shown LoRA to be effective for multi-task learning.
>
> While we do not think this intuition is enough to say that LoRA conditioning *should* be effective, we do believe it is sufficient to warrant a thorough investigation, and we empirically establish the effectiveness.
>
> [1] Christopher Williams, Fabian Falck, George Deligiannidis, Christopher C. Holmes, Arnaud Doucet, and Saifuddin Syed. A unified framework for u-net design and analysis. NeurIPS, 2023.

---

### Review · Reviewer_Rtei · 2024-07-03

**Summary Of Contributions:**

This paper suggests that LoRA blocks should be integrated within a U-net block as a part of a conditioning process. The authors show that architectural change leads to a small improved performance w.r.t the FID metric, at the cost of some additional parameters.

**Audience:**

Yes

**Broader Impact Concerns:**

None to note.

**Claims And Evidence:**

Yes

**Requested Changes:**

I think the paper is overall well written and the figures are nice. I was not particularly knowledgeable about this topic, but the paper did a good enough job of helping me understand the topic. I would say there could be some more information in the introduction about the theoretical motivation behind the paper.

**Strengths And Weaknesses:**

The fact that this is a drop-in improvement makes is a strong point, since otherwise, with architectural improvement, it can be difficult to tell if there is a meaningful improvement if there is significant hyper-parameter tuning involved.

---

> ### Author Response · Authors · 2024-07-29
> **Response to Reviewer Rtei**
>
> We thank the reviewer for the constructive feedback on the drop-in improvement and the presentation of our work.
>
> # Theoretical motivation (shared with response to Reviewer XaAm)
>
> As the reviewer points out, our work does not present a theoretical motivation behind our proposal. The lack of theoretical justification of the neural network architecture is a shared shortcoming of essentially all prior work in diffusion models. (The only theoretical work that we are aware of is [1], which provides post hoc justification.) Nevertheless, we did have some empirical motivation behind using LoRA for conditioning diffusion models:
>
> - Prior work has shown LoRA to be compatible with diffusion models for the purpose of fine-tuning.
> - The conditioning of the U-Net to learn the time- and class-dependent score function can be viewed as a multi-task learning setup, and prior work has shown LoRA to be effective for multi-task learning.
>
> While we do not think this intuition is enough to say that LoRA conditioning *should* be effective, we do believe it is sufficient to warrant a thorough investigation, and we empirically establish the effectiveness.
>
>
> [1] Christopher Williams, Fabian Falck, George Deligiannidis, Christopher C. Holmes, Arnaud Doucet, and Saifuddin Syed. A unified framework for u-net design and analysis. NeurIPS, 2023.

---

### Review · Reviewer_f6Uc · 2024-07-16

**Summary Of Contributions:**

This work examines the problem of including scalar/vector conditional information such as timestep and class label in the attention layers of UNet-based diffusion models. The majority of current methods include this type of conditional information only in the residual layer of the UNet, which seems suboptimal. The paper proposes to use LoRA to adapt the input QKV and output projection layers of UNet attention modules, with the motivation that this provides a lightweight and constrained adjustment to the operation basic attention layer. The major contribution of the work is designing two types of LoRA layers: one for discrete ordered timesteps or classes, and the other for continuous timesteps and continuous conditions. Experiments are conducted showing that including the proposed modules provides improvement in generative quality metrics.

**Audience:**

Yes

**Broader Impact Concerns:**

Broader impacts were not discussed, but that does not influence my view of the paper.

**Claims And Evidence:**

Yes

**Requested Changes:**

I have no major requested changes. It would be interesting to see whether UC-LoRA can serve as a single solution for all the cases considered, but this is not essential.

**Strengths And Weaknesses:**

Strengths:
* The aim of the paper is straightforward and clear. Including timestep information in the attention layers of diffusion-based UNets seems to be a missing component of the current paradigm and it is useful to explore ways to do so.
* The proposed LoRA method is efficient and aligns nicely with existing diffusion techniques. The handling of LoRA module design makes intuitive sense and seems natural.
* The experiments are conducted in a clear and reproducible way. There is solid evidence that consistent (although modest) refinement can be achieved by including the proposed modules.

Weaknesses:
* The introduction of two different LoRAs techniques for two different cases seems somewhat unnecessary. Discrete timesteps can be treated as continuous inputs, and classes can be represented as vectors in one-hot or binary encoding, etc. which can also be treated as continuous inputs. I would be interested in seeing evidence whether UC-LoRA can match Time and Class LoRA in the cases where Time and Class LoRA were used. If so, it might be more straightforward to adopt a single approach. This is not a major concern, but it is worth exploring.
* I am not sure how interpolation between classes would be performed for Class LoRA in the case of a large number of classes. The interpolation method for timesteps seems to depend on an ordering of the timesteps, which doesn't hold for classes. I would be interested in hearing how handling a large number of classes could be done.

---

> ### Author Response · Authors · 2024-07-29
> **Response to Reviewer f6Uc**
>
> We thank the reviewer for the constructive feedback. We are happy to hear that the reviewer found our proposed method efficient, sound, and intuitive.
>
> # Using UC-LoRA for Time and Class LoRA
>
> As the reviewer points out, UC-LoRA is capable of handling both discrete and continuous inputs. Indeed in the EDM experiments, UC-LoRA receives both continuous and discrete inputs, the SNR level and class label, respectively. Nevertheless, we believe that there are several advantages of (discrete) Time and Class LoRA that make them worth considering:
>
> - The architectural choice is straightforward (whereas for UC-LoRA, optimal architecture for MLP block requires further architectural exploration and fine-tuning).
> - In the case of TimeLoRA, we can directly impose the task similarity between nearby timesteps at initialization, hence enhancing stability during the training
>
> # Compositional ClassLoRA
>
> The interpolation between classes, as the reviewer discusses, would be achieved with a *compositional* version of ClassLoRA. This would indeed be necessary when the number of classes becomes very large. As the reviewer points out, task similarity between classes are not straightforward, however, we still believe that the compositional ClassLoRA can handle large number of classes.
>
> To verify the effectiveness of a compositional version of ClassLoRA, we conducted a proof-of-concept experiment with EDM on CIFAR-10. We finetuned a pretrained unconditional EDM for class-conditional CIFAR-10 sampling using a *single* LoRA basis and randomly initialized compositional weights. Note in the non-compositional ClassLoRA, we used 10 LoRA bases, one for each class. As a result, with the addition of only 61560 parameter counts, we were able to convert (fine-tune) an unconditional EDM to a conditional EDM and improve the FID score from 1.97 to 1.81. We have added this experiment in Appendix B of our manuscript.

---

### Decision · Action_Editor_DKY1 · 2024-08-29

**Recommendation:** Accept as is

**Comment:**

The paper proposes a new technique of including conditioning in the parameterization of diffusion models using LoRA for UNets. The paper has a clear objective and well-defined contributions for which enough empirical results are presented.

The main strengths of the paper are the following:
- The paper has a clear goal of injecting timestep information in attention layers of UNet-based diffusion models to improve the model ability to adapt to different denoising timesteps and improve sampling results.
- The proposed LoRA method is efficient.
- The experiments are conducted in a clear and reproducible way. There is solid evidence that consistent (although modest) refinement can be achieved by including the proposed modules.

There are no major weaknesses.

All reviewers vote to accept the paper. I agree with them and recommend accepting the paper.

**Audience:**

The paper is definitely interesting for the TMLR's audience. The class of diffusion-based models is a leading collection of generative models nowadays. Moreover, LoRA is a technique that is widely used in the industry.

**Claims And Evidence:**

The paper proposes to use LoRA to condition time and class information on attention layers in UNet-based diffusion models. The main claims are the following:
(i) The proposed method improves image generation measured by FID;
(ii) The paper identifies how to condition UNets on time and class information.

The claims are properly stated and they constitute interesting research objectives. The reviewers and I see enough evidence for these claims.